# Scaling Offline Model-Based RL via Jointly-Optimized World-Action Model Pretraining

**Jie Cheng**[1,2]**, Ruixi Qiao**[1,2]**, Yingwei Ma**[3]**, Binhua Li**[3]**,**
**Gang Xiong**[1,2]**, Qinghai Miao**[2]**, Yongbin Li**[3*]**, Yisheng Lv**[1,2*]
[1]State Key Laboratory of Multimodal Artificial Intelligence Systems,
  Institute of Automation, Chinese Academy of Sciences
[2]School of Artificial Intelligence, University of Chinese Academy of Sciences
[3]Alibaba Group

## Abstract

A significant aspiration of offline reinforcement learning (RL) is to develop a generalist agent with high capabilities from large and heterogeneous datasets. However, prior approaches that scale offline RL either rely heavily on expert trajectories or struggle to generalize to diverse unseen tasks. Inspired by the excellent generalization of world model in conditional video generation, we explore the potential of image observation-based world model for scaling offline RL and enhancing generalization on novel tasks. In this paper, we introduce JOWA: **J**ointly-**O**ptimized **W**orld-**A**ction model, an offline model-based RL agent pretrained on multiple Atari games with 6 billion tokens data to learn general-purpose representation and decision-making ability. Our method jointly optimizes a world-action model through a shared transformer backbone, which stabilize temporal difference learning with large models during pretraining. Moreover, we propose a provably efficient and parallelizable planning algorithm to compensate for the Q-value estimation error and thus search out better policies. Experimental results indicate that our largest agent, with 150 million parameters, achieves 78.9% human-level performance on pretrained games using only 10% subsampled offline data, outperforming existing state-of-the-art large-scale offline RL baselines by 71.4% on averange. Furthermore, JOWA scales favorably with model capacity and can sample-efficiently transfer to novel games using only 5k offline fine-tuning data (approximately 4 trajectories) per game, demonstrating superior generalization. The code and checkpoints will be released at https://github.com/CJReinforce/JOWA.

## 1 Introduction

In recent years, building large-scale generalist models capable of solving multiple tasks has become a dominant research focus in natural language processing (NLP) and multi-modality (Yue et al., 2024). Their success is largely driven by the scaling law (Kaplan et al., 2020), which posits that increasing model size and data leads to improved performance. However, similar scaling trends have not been extensively observed in reinforcement learning (RL).

Unlike in vision and language domains, RL has traditionally favored smaller models tailored to single tasks or multiple tasks within the same environment. Concerningly, previous studies have shown that scaling model capacity can lead to instabilities or performance degradation (Kumar et al., 2020a; Ota et al., 2021; Sokar et al., 2023). While some efforts have been made to scale offline RL across multiple tasks (Lee et al., 2022; Xu et al., 2022), they predominantly rely on supervised learning (SL) approaches, such as conditional behavior cloning, rather than temporal difference (TD) learning, and heavily rely on large amounts of expert trajectories. Kumar et al. (2023) scaled offline Q-learning using ResNet-based representation network with separate Q-heads for each game, but this approach only learns generalizable representations and still requires substantial data and gradient steps to adapt to unseen games due to reset of Q-heads during fine-tuning. Therefore,

---

*Equal corresponding author

*scaling TD-based offline RL for simultaneous **general-purpose representation and decision-making** remains a critical challenge.* While Hansen et al. (2024) attempted to address this challenge for continuous control tasks with low-dimensional proprioceptive states using model-based RL, it lacks generalization due to heterogeneous proprioceptors across task domains.

Meanwhile, world model has decoupled from model-based RL and evolved into a distinct research area within computer vision and multi-modality, primarily focusing on conditional video generation (Hong et al., 2022; Blattmann et al., 2023; Yu et al., 2023; Yang et al., 2024; Bruce et al., 2024). Notably, SORA (Brooks et al., 2024) has demonstrated superior generalization performance as a world simulator through large-scale training of generative models on time-series image data. This motivates our investigation into a compelling question: *"Can image observation-based world model scale offline RL across multiple tasks while enhancing generalization to diverse unseen tasks?"*

To address this question, we introduce JOWA: **J**ointly-**O**ptimized **W**orld-**A**ction model, an offline model-based RL agent pretrained across multiple visual games with approximately 6 billion tokens data. Crucially, JOWA unlocks scaling trends and achieves sample-efficient adaption to novel games. By utilizing a shared transformer backbone for both the world model and Q-value criticism, JOWA learns both generalizable representations and decision-making skills. This architecture allows the transformer to absorb gradients back-propagated from both world modeling and TD losses, enabling joint optimization. The world modeling loss acts as a regularizer, stabilizing TD-learning for large models. Additionally, we propose a provably efficient and parallelizable planning algorithm to compensate for the Q-value estimation error, allowing for consistent identification of the optimal policy at inference time and sample-efficient transfer to novel games.

To evaluate the performance of JOWA, we train a single model to play 15 Atari games, similar to Lee et al. (2022) but using a reduced yet sufficient dataset of 10M transitions per game—termed as the low-data regime to highlight the data efficiency. This setup presents a significant challenge due to the unique dynamics, visuals, and agent embodiments of games. To further test JOWA's generalization, we perform offline fine-tuning on 5 unseen games, using minimal fine-tuning data.

Our contributions are threefold: First, we introduce JOWA, an offline model-based RL method capable of training a single high-performing generalist agent across multiple Atari games. JOWA attains 78.9% human-level performance on pretrained games using only 10% of the original dataset (Agarwal et al., 2020), outperforming existing state-of-the-art large-scale offline RL baselines by 71.4% on averange. Second, we demonstrate that JOWA unlocks scaling trends, with performance improving as model capacity increases. Third, JOWA enables sample-efficient transfer to diverse unseen games with 64.7% DQN-normalized score using only 5k transitions per game, surpassing baselines by 69.9% on average. Our ablation studies highlight the significance of two key design features of JOWA: joint optimization and planning, along with other training choices. We release all the training, evaluation, and fine-tuning code, as well as model weights, to support future research.

## 2 RELATED WORK

**Offline Reinforcement Learning.** Offline RL algorithms learn a policy entirely from the static offline dataset without online interactions. Model-free offline RL incorporates conservatism to mitigate extrapolation error (Jin et al., 2021) primarily through policy constraints (Fujimoto et al., 2019; Kumar et al., 2019; Fujimoto & Gu, 2021; Cheng et al., 2024; Liu et al., 2024; 2025) and value regularization (Kumar et al., 2020b). Model-based offline RL approximates the environment using world models and performs conservative policy optimization (Yu et al., 2020; 2021). While these works focus on single-task settings, our work explores scaling offline model-based RL across diverse, challenging multi-task Atari games (Lee et al., 2022; Kumar et al., 2023; Wu et al., 2024) aiming for sample-efficient transfer to novel games.

**Mutli-Task Reinforcement Learning.** Multi-task reinforcement learning (MTRL) aims to learn a shared policy for diverse tasks. A common approach is to formulate the multi-task model as task-condition, such as language-conditioned tasks (Ahn et al., 2022; Jang et al., 2022) and goal-conditional RL (Plappert et al., 2018). In multi-task offline RL, conditional sequence modeling approaches based on decision transformer (Chen et al., 2021) or diffusion model (Janner et al., 2022) typically rely on large amounts of expert trajectories (Xu et al., 2022; Lee et al., 2022; Wu et al., 2024; Hu et al., 2024; He et al., 2024). Beyond that, FICC (Ye et al., 2022) pretrains multi-task representation and dynamic models with action-free videos and then fine-tunes a model-based

agent on each task for fast adaptation. Scaled-QL (Kumar et al., 2023) scales offline Q-learning using a shared feature network across tasks with separate Q-value heads for each task. Our work advances offline TD learning to multi-task settings without task-specific Q-value heads through a jointly-optimized world-action model. Table 1 compares experimental environments and open-source status of multi-task offline RL algorithms. Following Lee et al. (2022); Kumar et al. (2023); Wu et al. (2024), we focus on the multi-game regime, which presents greater challenges due to high-dimensional observations and diverse, stochastic environment dynamics.

Table 1: Comparison of methods in multi-task offline RL. ♣ and ♠ represent two training paradigms of agents, conditional BC and TD-learning, respectively.

| Method | Experimental environment | | | | | Open source | | |
| | Benchmark | Observation space | Action space | Dynamics | | Train | Eval | Check-points |
| | | | | per task | across tasks | | | |
| HarmoDT♣ (Hu et al., 2024) | Meta-World or DMControl | state | continous | deterministic | same or similar | ✓ | ✓ | ✗ |
| TD-MPC2♠ (Hansen et al., 2024) | | | | | | ✓ | ✓ | ✓ |
| FICC♠ (Ye et al., 2022)[1] | Atari 2600 | pixels | discrete | stochastic | diverse | ✗ | ✗ | ✗ |
| MGDT♣ (Lee et al., 2022) | | | | | | ✗ | ✓ | ✓ |
| Elastic DT♣ (Wu et al., 2024) | | | | | | ✓ | ✓ | ✗ |
| Scaled-QL♠ (Kumar et al., 2023)[2] | | | | | | ✗ | ✗ | ✗ |
| JOWA (ours)♠ | | | | | | ✓ | ✓ | ✓ |

## 3 PRELIMINARIES AND PROBLEM SETUP

### 3.1 ONLINE DISTRIBUTIONAL RL (C51)

In distributional RL, the distribution $Z$ over returns replaces the Q-value in the Bellman optimality equation. The Q-value is the mean of the value distribution $Z$ that can be computed through a distributional Bellman optimality operator (Bellemare et al., 2017),

$$\mathcal{T}^* Z(s, a) :\overset{D}{=} R(s, a) + \gamma Z(s', \arg\max_{a'} Q(s', a')) \tag{1}$$

where the formula $Y :\overset{D}{=} U$ denotes equality of probability laws, that is the random variable $Y$ is distributed according to the same law as $U$. The C51 algorithm (Bellemare et al., 2017) models $Z(s, a)$ using a discrete distribution supported on $N$ fixed atoms $z_1 \leq \cdots \leq z_N$ uniformly spaced over a predetermined interval. Given a current value distribution, C51 applies a projection step to map the target distribution onto its finite element support and optimizes as follows:

$$\mathcal{L}_{\text{TD}} = D_{\text{KL}}(\mathcal{T}^* Z_{\theta^-}(s, a) \| Z_\theta(s, a)) \tag{2}$$

### 3.2 VALUE REGULARIZATION BASED OFFLINE RL (CQL)

To be conservatism on unseen actions, CQL (Kumar et al., 2020b) introduces a regularizer to the TD-loss, which minimizes Q-values for unseen actions while maximizing Q-values for actions in the dataset to counteract excessive underestimation. The loss function for CQL is given by:

$$\mathcal{L}_{\text{CQL}} = \alpha \left( \mathbb{E}_{s \sim \mathcal{D}} \left[ \log \left( \sum_{a'} \exp(Q_\theta(s, a')) \right) \right] - \mathbb{E}_{s,a \sim \mathcal{D}} [Q_\theta(s, a)] \right) + \mathcal{L}_{\text{TD}} \tag{3}$$

### 3.3 PROBLEM SETUP

We consider a multi-task offline RL problem: given a static dataset of transitions $\mathcal{D} = \{(s_t, a_t, r_t, d_t, s_{t+1})_i\}$ collected from various environments with arbitrary behaviour polices, our goal is to learn a single policy that maximizes the expected return $\mathcal{R}_t = \sum_{k \geq t} \gamma^{k-t} r_k$ on all considered environments and can be efficiently fine-tuned to new tasks, where $\gamma$ is the discount factor.

Considering the computational budget, we use 15 Atari games for pretraining and 5 games for out-of-distribution (OOD) experiments. The whole training process took around 12 days on A100 GPUs.

---

[1]FICC only open-sourced the world model pretraining code without the agent fine-tuning code.

[2]Scaled-QL released the preliminary code which is not run-able out-of-the-box.

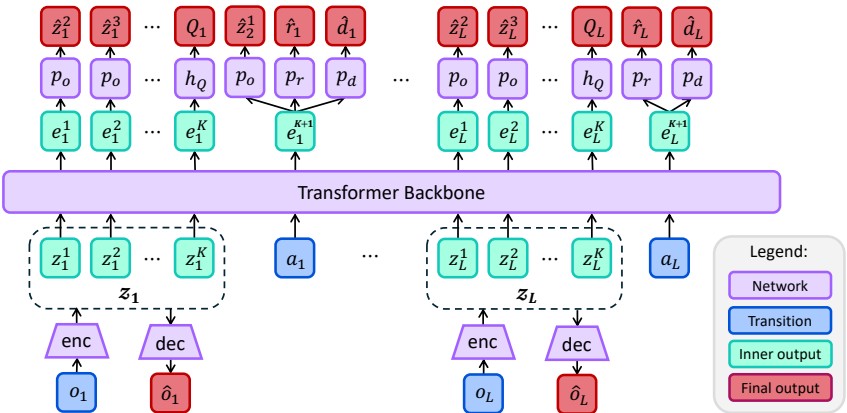

Figure 1: Architecture of JOWA. We use a shared transformer backbone for both world modeling and Q-value criticism to enable joint optimization. VQ-VAE tokenizes images into visual tokens. The sum of vocabulary embeddings, position embeddings and task embeddings forms the input embeddings space for the transformer backbone.

Our offline dataset is derived from the the DQN-Replay dataset (Agarwal et al., 2020), which consists of $84 \times 84$ grayscale images as observations and a full action space with 18 discrete actions.

## 4 JOINTLY-OPTIMIZED WORLD-ACTION MODEL

In this section, we first detail the architecture of JOWA and the loss functions for joint optimization in section 4.1. Next, we introduce the provably efficient and parallelizable planning algorithm employed to compensate for the Q-value estimation error in section 4.2. Finally, we present the overall training pipeline in section 4.3.

### 4.1 WORLD-ACTION MODEL

#### 4.1.1 ARCHITECTURE

Figure 1 illustrates JOWA's architecture, which uses a transformer backbone (Vaswani, 2017) to simultaneously learn world dynamics and Q-values across environments. This dual capability is achieved through distinct prediction heads that process the transformer's output embedding $e$. The world dynamics are modeled via supervised learning using three heads: next observation token predictor $p_o$, reward predictor $p_r$, and termination predictor $p_d$. The Q-values head $h_Q$ learns the Q-function through TD-learning, based on implicit representations of historical trajectories. In the following sections, we refer to the {transformer, $p_o$, $p_r$, $p_d$} components as the "world-part" with parameters $\theta$, and the {transformer, $h_Q$} components as the "action-part" with parameters $\phi$.

We use VQ-VAE, a discrete autoencoder, to tokenize image observations, representing high-dimensional images as a sequence of $K$ tokens. The VQ-VAE is trained with an equally weighted combination of $L_1$ reconstruction loss, commitment loss (Van Den Oord et al., 2017), and perceptual loss (Esser et al., 2021). Then the transformer operates on the interleaved observation and action tokens represented as $(z_0^1, \ldots, z_0^K, a_0, \ldots, z_L^1, \ldots, z_L^K, a_L)$, where $L$ is the maximum timesteps of the trajectory segments. For multi-task learning, we incorporate learnable task embeddings for both observation and action tokens.

#### 4.1.2 TRAINING OF WORLD-PART MODULE

At each timestep $t$, the world-part module models the following distributions:

Dynamics predictor: $\quad \hat{z}_t^k \sim p_o\big(\hat{z}_t^k \mid f(z_{\leq t-1}, z_t^{<k}, a_{\leq t-1}, u)\big)$ (4)

Reward predictor: $\quad \hat{r}_t \sim p_r\big(\hat{r}_t \mid f(z_{\leq t}, a_{\leq t}, u)\big)$ (5)

Termination predictor: $\hat{d}_t \sim p_d\big(\hat{d}_t \mid f(z_{\leq t}, a_{\leq t}, u)\big)$ (6)

where $f$ represents the transformer backbone, $u$ is the task ID to index the corresponding task embedding, $k \in \{1, \cdots, K\}$, and $t \in \{1, \cdots, L\}$.

To unify the category of loss functions for balanced training (Vandenhende et al., 2021), we convert scalar rewards to ternary quantities $\{-1, 0, 1\}$ using the sign function. This allows all three predictors to be optimized as classification problems by cross-entropy loss. Given $L$-timesteps segments sampled from the offline dataset, the loss function for the world-part module is formulated as:

$$\mathcal{L}_{\text{world}}(\theta) = \frac{1}{L} \sum_{t=1}^{L} \Big[ \frac{1}{K} \sum_{k=1}^{K} - \ln p_o \big( z_t^k \mid f(z_{\leq t-1}, z_t^{<k}, a_{\leq t-1}, u) \big)$$
$$- \ln p_r \big( r_t \mid f(z_{\leq t}, a_{\leq t}, u) \big) - \ln p_d \big( d_t \mid f(z_{\leq t}, a_{\leq t}, u) \big) \Big] \tag{7}$$

### 4.1.3 TRAINING OF ACTION-PART MODULE

We use CQL for offline TD-learning during pretraining. To ensure training stability and enhance scaling performance (Farebrother et al., 2024), we employ distributional TD-error (Bellemare et al., 2017) instead of the mean-square TD-error, maintaining consistency with the $\mathcal{L}_{\text{world}}$ loss category.

The action-part module computes the return distribution for all actions given an observation and historical information. The value function $Q(o_t, a_t)$ is the mean of $\mathcal{Z}(o_t, a_t)$. Then the loss function for the action-part module is formulated as:

$$\mathcal{L}_{\text{action}}(\phi) = \alpha \left[ \log \left( \sum_a \exp \left( Q(o_t, a) \right) \right) - Q(o_t, a_t) \right] + D_{\text{KL}} \left( \mathcal{T}^* \mathcal{Z}^-(o_t, a_t) \| \mathcal{Z}(o_t, a_t) \right) \tag{8}$$

where $\mathcal{T}^*$ is the distributional Bellman optimality operator defined in Equation (1), and $\mathcal{Z}^-$ is the target distribution computed through a target Q-values head $h_{Q^-}$. We set $\alpha = 0.1$ in experiments.

Therefore, the joint optimization objective for the world-action model is formulated as:

$$\mathcal{L}(\theta, \phi) = \beta \mathcal{L}_{\text{world}}(\theta) + \mathcal{L}_{\text{action}}(\phi) \tag{9}$$

with the coefficient $\beta > 0$. We set $\beta = 0.1$ in our experiments.

Both $\mathcal{L}_{\text{world}}$ and $\mathcal{L}_{\text{action}}$ back-propagate gradients to the transformer. Previous works observed that TD methods suffer from greater instability with larger model size (Kumar et al., 2020a; Sokar et al., 2023). However, through jointly optimizing the world-action model, $\mathcal{L}_{\text{world}}$ serves as a regularizer to stabilize TD-learning in large-scale models. Moreover, the world-part module enables planning at decision time for optimal inference and sample-efficient transfer, detailed in the following sections.

## 4.2 PARALLELIZABLE PLANNING AT INFERENCE TIME

The world-part module enables planning at decision time to compensate for inaccurate Q-value estimates, allowing JOWA to consistently search out the optimal policy. We model this process as a tree search problem and present a practical, parallelizable search algorithm.

Given an estimated optimal Q-value $\hat{Q}^*$, a learned world model with dynamic predictor $\hat{P}$ and reward predictor $\hat{r}$, our objective is to find the optimal action $a_0^*$ maximizing the ground-truth optimal Q-value $Q^*$, starting from state $s_0$. To do so, we rewrite the Bellman optimality equation as:

$$Q^*(s_0, a_0) = \max_{\pi_{Q^*}} \mathbb{E}_{\substack{s_1, \cdots, s_H \sim P \\ a_1, \cdots, a_{H-1} \sim \pi_{Q^*}}} \left[ \sum_{t=0}^{H-1} \gamma^t r(s_t, a_t) + \gamma^H \max_{a_H} Q^*(s_H, a_H) \right] \tag{10}$$

where $\pi_{Q^*}$ is the policy induced by the optimal Q-function. For Equation (10), the optimal policy is the greedy policy based on $Q^*$. The proof is provided in the Appendix A.

To derive the search objective function, we follow three steps: (i) replace the ground-truth functions in the right side of Equation (10) with estimated or learned functions. (ii) leverage the estimated optimal Q-function $\hat{Q}^*$ to reduce the policy space for search, restricting the actions to those with top-$K$ highest Q-values. Denote the constrained policy space as $\Pi_{\hat{Q}^*}$, where $\forall \pi \in \Pi_{\hat{Q}^*}, \forall s \in \mathcal{S}, \forall a \notin \text{top-}K(\hat{Q}^*(s, \cdot))$, we have $\pi(a|s) = 0$. (iii) maximize over $a_0$ on both sides of Equation

(10) to find the optimal initial action, considering the restriction in the second step. Finally, the resulting objective function is formulated as:

$$\max_{\pi \in \Pi_{\hat{Q}^*}} \mathbb{E}_{\substack{s_1, \cdots, s_H \sim \hat{P} \\ a_0, \cdots, a_{H-1} \sim \pi}} \left[ \underbrace{\sum_{t=0}^{H-1} \gamma^t \hat{r}(s_t, a_t)}_{\text{income-to-date}} + \underbrace{\gamma^H \max_{a_H} \hat{Q}^*(s_H, a_H)}_{\text{income-to-go}} \right] \qquad (11)$$

Detailed derivation is provided in the Appendix B. Then we show the error bound of search-based optimal Q-function in Theorem 4.1, with the formal theorem and its proof in Appendix C.

**Theorem (Informal) 4.1.** *Denote the search-based optimal Q-function as $f(s, a)$. Assume the learned reward function $\hat{r}$ to be $L_r$-Lipschitz and the estimated optimal Q-function $\hat{Q}^*$ to be $L_Q$-Lipschitz. Assume the estimation errors of the learned state transition, reward, and Q-value are bounded by $\epsilon_s, \epsilon_r, \epsilon_Q$ respectively. Then we have the error between search-based optimal Q-value $f(s, a)$ and ground-truth optimal Q-value $Q^*(s, a)$ bounded as:*

$$\|f(s, a) - Q^*(s, a)\| \leq \frac{1 - \gamma^H}{1 - \gamma} \epsilon_r + \left( \frac{\gamma - \gamma^H}{1 - \gamma} \epsilon_r + \gamma^H \epsilon_Q \right) \epsilon_s + \gamma^H \epsilon_Q \qquad (12)$$

*Under the following condition, the search-based optimal Q-function $f$ has an upper error bound no greater than the estimated optimal Q-function $\hat{Q}^*$:*

$$\frac{1}{1 - \gamma} \epsilon_r + \left( \frac{\gamma}{1 - \gamma} \frac{L_r - \gamma^H L_Q}{1 - \gamma^H} - \frac{L_r - L_Q}{1 - \gamma} \frac{\gamma^H}{1 - \gamma^H} \right) \epsilon_s \leq \epsilon_Q \qquad (13)$$

To optimize objective function (11), we interact with the imagined MDP induced by the learned world model using the constrained policy $\pi \in \Pi_{\hat{Q}^*}$ for $H$ steps, starting from $s_0$. We then compute the total income (income-to-date plus income-to-go) for all leaf nodes to identify the optimal path. The first edge of this path is the optimal initial action.

We implement this search using beam search, an efficient decoding algorithm common in NLP. At each timestep, we retain only $K$ states with the top-$K$ total income values. The horizon $H$ and beam width $K$ are hyper-parameters, with $K = 1$ or $H = 0$ degenerating to a $\hat{Q}^*$-based greedy policy. The algorithm calls the world model $K^2(H - 1) + K$ times but only takes $H$ times as long as the forward propagation of the world model due to parallelizability across $K$ beams.

### 4.3 TRAINING PIPELINE

Our multi-task offline RL pretraining consists of two stages:

- **Stage 1:** Sample trajectory segments from datasets. Train the VQ-VAE tokenizer using image observations. Train the world-part module using segments with loss (7) for $M_1$ steps.
- **Stage 2:** Freeze the VQ-VAE tokenizer. Sample segments and jointly optimize the world-action model with loss function (9) for $M_2$ steps.

We employ this two-stage training approach to stabilize and accelerate the overall training process. In our pretraining experiments, we set $M_1 = 250\text{k}$ and $M_2 = 1.5\text{M}$, totaling $1.75\text{M}$ gradient steps.

## 5 EXPERIMENTS

We design our experiments to answer the following questions: **(1)** How does JOWA perform on multi-games in low-data regime? **(2)** Can JOWA effectively leverage higher model capacity? **(3)** Does the pre-trained JOWA sample-efficiently transfer to new games?

## 5.1 Experimental Setup

**Dataset.** We use the Atari dataset from Agarwal et al. (2020), which contains 50M transitions from each of 5 separate training runs. Due to the prohibitive long training time on full games, we select a subset of 20 games, maintaining the difficulty distribution of full games defined by Gulcehre et al. (2020). These 20 games are introduced in Appendix D. 15 of those games are used for training, and 5 games are held out for OOD generalization experiments. Following Lee et al. (2022), we use data from 2 out of 5 training runs. To investigate performance in low-data regime, we uniformly draw 10% of transitions at random, as per Agarwal et al. (2020), resulting in 10M transitions per game.

**Training and Fine-tuning.** We implement our world-action model based on GPT-2 (Brown et al., 2020). We train three JOWA variants: JOWA-150M (150M parameters), JOWA-70M, and JOWA-40M. We set the number of visual tokens $K$ to 36, resulting in a total of approximately 6B tokens in our dataset. The sequence length $L$ is set to 8. We pretrain all JOWA models on A100 GPUs for 1.75M steps. For fine-tuning, we train models for 50k gradient steps with 5k transitions.

**Evaluation and Metrics.** During evaluation, we enable planning for JOWA, setting the planning horizon $H$ to 2 for all games and adjust the beam width $K$ based on the valid action space size of each game. See Appendix E.3 for the detailed evaluation protocol. We measure performance using human normalized scores (HNS) (Mnih et al., 2015), i.e. $(\text{score} - \text{score}_{\text{random}})/(\text{score}_{\text{human}} - \text{score}_{\text{random}})$. To create an aggregate comparison metric across all games, we use inter-quartile mean (IQM) of human-normalized scores, following evaluation best practices proposed in (Agarwal et al., 2021).

More details on hyperparameters, network architecture, algorithm implementation, and protocols for fine-tuning, and evaluation are provided in the Appendix E.

## 5.2 Baseline Methods

We compare JOWA with the following baselines: (i) **Multi-Task BC (MTBC).** Our method can naturally be reduced to a transformer-based multi-task behavioral cloning agent. We train MTBC models at 3 scales: 34M, 65M, 120M parameters. (ii) **Multi-Game DT (MGDT).** Lee et al. (2022) used $\{o_{t+i}, \hat{R}_{t+i}, a_{t+i}, r_{t+i}\}_{i=0}^{3}$ as input sequences, where $\hat{R}$ is the target return, and expert-conditioned return distribution to induce expert-level actions during evaluation. We train MGDT models at 2 scales: 40M and 200M parameters. (iii) **Elastic DT (EDT).** Based on MGDT, Wu et al. (2024) removed the reward in the input sequences and dynamically selected history length to address challenges of trajectory stitching. We train EDT based on the architecture configuration of MGDT-200M. (iv) **Scaled-QL (SQL).** Kumar et al. (2023) scaled offline Q-learning through pre-trained and shared representation network and separate Q-values head for each game. We reproduce the largest Scaled-QL with 80M parameters using ResNet-101 network. (v) **FICC.** Ye et al. (2022) pretrained the dynamic model with videos and then online fine-tuned the agent on each game. To obtain an offline multi-task policy, we pretrain FICC for 0.5M steps and then primarily fine-tune the $Q$-function on 15 games all at once, the same as JOWA's second pretraining stage, for 1.25M steps. We replace all residual blocks in FICC's official code with ResNet-50-style blocks, resulting in FICC-85M. For fair comparison, all methods use the same batch size of 512 and 1.75M gradient steps. The implementation details of all baselines are provided in the Appendix E.1.

## 5.3 How does JOWA perform on multi-games in low-data regime?

We summarize our main results in Table 2. This table shows the performance of JOWA alongside all best performing sizes of baselines trained with 10% subsampled dataset. MTBC, MGDT, and EDT represent (conditional) behavior cloning methods, while Scaled-QL, FICC, and JOWA represent Q-learning methods. Despite the constraints on the amount of data, JOWA achieves superior performance with comparable or fewer parameters than baselines. Specifically, JOWA-150M attains an IQM human-normalized score of 78.9% in low-data regime, surpassing MGDT-200M (49.8%) and EDT-200M (50.2%), despite these two models having more parameters and utilizing data augmentation during pretraining. JOWA-70M achieves an IQM human-normalized score of 47.6%, outperforming Scaled-QL-80M (42.0%) and FICC-85M (43.3%). JOWA maintains its performance advantage when comparing median human-normalized scores. These results demonstrate JOWA's sample efficiency in learning from heterogeneous offline data.

Table 2: Returns on the 15 Atari games for best performing sizes of multi-game models trained with the 10% subsampled dataset. Bold numbers indicate the top methods.

| Game | Random | Human | MTBC 120M | MGDT 200M | EDT 200M | SQL 80M | FICC 85M | JOWA 40M | JOWA 70M | JOWA 150M |
|---|---|---|---|---|---|---|---|---|---|---|
| Assault | 222.4 | 742.0 | 1203.5 | 1741.5 | 1915.5 | 2292 | 925.9 | 1578.3 | 1733.9 | **2302** |
| Atlantis | 12850 | 29028.1 | 37812.5 | 2565750 | 2108166.7 | 49100 | 86250 | 41662.5 | 570862.5 | **2690387.5** |
| BeamRider | 363.9 | 16926.5 | 786 | 6011.3 | 4149.8 | **7023.5** | 6822 | 880.3 | 2547.4 | 3498 |
| Berzerk | 123.7 | 2630.4 | 655 | 444.5 | 350 | 312.5 | 400 | 395 | 441.9 | **739** |
| Carnival | 0 | 3800 | **5560** | 2610 | 3678.8 | 1940 | 2820 | 4685 | 4070 | 5316 |
| Centipede | 2090.9 | 12017 | 4046.6 | 4604 | 3389.4 | 3650.3 | 3742.2 | **5669.3** | 4475.6 | 4677 |
| ChopperCommand | 811.0 | 7387.8 | 256.3 | 3300.8 | 3443.8 | 853.8 | 2835.6 | 3118.8 | 2568.8 | **3812.5** |
| DemonAttack | 152.1 | 1971 | 2611.3 | 6549.4 | 3455.7 | **7936.5** | 5806.4 | 1233.8 | 4584.4 | 3547.8 |
| NameThisGame | 2292.3 | 8049 | 6045 | 6610.5 | 7060 | 6461.7 | 6236 | 7335.6 | **12706.9** | 11421 |
| Phoenix | 761.4 | 7242.6 | 1446.9 | 5120.5 | 5320 | 4961.7 | 3814.5 | 1608.8 | 5065 | **5348** |
| Seaquest | 68.4 | 42054.7 | 120 | 2720 | **3160.4** | 650 | 1760 | 1033.1 | 1490 | 2725 |
| SpaceInvaders | 148 | 1668.7 | 605 | 742.5 | 513.8 | 714.4 | 641.2 | 694.7 | **969.1** | 744.7 |
| StarGunner | 664.0 | 10250 | 1493.8 | 8625.5 | 9550 | 4728.6 | 4936.4 | 5737.5 | **21231.3** | 18150 |
| TimePilot | 3568 | 5229.2 | 762.5 | 3866.7 | 2812.5 | 4072.7 | **4166.7** | 3268.8 | 3831.3 | 3669 |
| Zaxxon | 32.5 | 9173.3 | 0 | 462.5 | 325.5 | 0 | 312.5 | 0 | 225 | **2163** |
| #Superhuman | 0 | N/A | 4 | 3 | 4 | 3 | 3 | 3 | 4 | **6** |
| Median HNS | 0.000 | 1.000 | 0.197 | 0.391 | 0.400 | 0.387 | 0.390 | 0.360 | 0.390 | **0.456** |
| IQM HNS | 0.000 | 1.000 | 0.329 | 0.498 | 0.502 | 0.420 | 0.433 | 0.371 | 0.476 | **0.789** |

Next, we present experiments highlighting JOWA's scaling and generalization capabilities. These two key properties derived from multi-game pretraining align with recent advancements in SL.

### 5.4 HOW DOES JOWA SCALES WITH MODEL SIZE?

Scaling law depicts the positive correlation between model capacity and performance. Following Lee et al. (2022); Kumar et al. (2023), we investigate JOWA's ability to leverage higher capacity architectures. For comparison of scaling trends, we establish 2 baselines: (i) MTBC scaled to 34M, 65M and 120M parameters. (ii) MGDT with 40M and 200M parameters. We scale JOWA by increasing the size of the transformer backbone and Q-values heads, resulting in 3 variants with 40M, 70M, and 150M parameters. Figure 2 shows the scaling trends for the 3 algorithms, which demonstrates that JOWA's performance reliably increases as the model size grows. Although previous works observed that TD-learning suffer from greater instability with larger model size (Lee et al., 2022; Sokar et al., 2023), JOWA scales TD-learning through a scalable transformer architecture and an auxil-

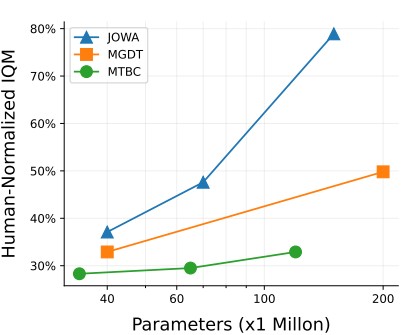

Figure 2: Scaling trends for different algorithms on the training set games.

iary regularization loss $\mathcal{L}_{\text{world}}$ to stabilize the TD-learning in large models. Notably, JOWA exhibits the steepest scaling curve among all algorithms, highlighting the great scalability potential of offline model-based RL.

Table 3: Fine-tuning performance on unseen games using 5k transitions, measured in terms of DQN-normalized score, following Lee et al. (2022). See Table 18 in Appendix F for raw scores.

| DNS | MTBC 120M | MGDT 200M | EDT 200M | SQL 80M | FICC 85M | JOWA 40M | JOWA 70M | JOWA 150M | JOWA-150M (scratch) |
|---|---|---|---|---|---|---|---|---|---|
| Mean | 0.164 | 0.422 | 0.430 | 0.360 | 0.543 | 0.504 | 0.576 | **0.647** | 0.196 |
| Median | 0.215 | 0.354 | 0.325 | 0.284 | 0.565 | 0.512 | **0.715** | 0.615 | 0.173 |
| IQM | 0.205 | 0.377 | 0.380 | 0.355 | 0.575 | 0.498 | 0.603 | **0.647** | 0.181 |

### 5.5 CAN JOWA SAMPLE-EFFICIENTLY TRANSFER TO NEW GAMES?

Rapid adaptation to downstream tasks is a natural and well-motivated benefit of pretraining. To investigate this question, we fine-tune pretrained agents on 5 held-out games using uniformly subsampled 5k expert-level transitions (from last 20% of DQN-Replay (Agarwal et al., 2020)) per game as the benchmark. These tiny amounts of transitions corresponding to approximately 4 trajectories from expert-level DQN-Replay per fine-tuned game on average, which is similar to the settings of

few-shot learning and is extremely challenging. For comparison, we fine-tune the largest agents of baselines alongside a no-pretrained baseline, JOWA-150M (scratch) trained on each held-out game from scratch. All pretrained agents and JOWA-150M (scratch) are fine-tuned for 50k and 500k gradient steps, respectively. Detailed fine-tuning protocol is shown in Appendix E.2. We report results in terms of DQN-normalized score (DNS), following Lee et al. (2022), in Table 3.

The results show that the fine-tuned JOWA-150M attains 64.7% IQM DNS across 5 held-out games, outperforming baselines by 69.9% on average. Moreover, comparing the performance of 3 JOWA variants, the scaling trend on IQM DNS still holds for novel games after fine-tuning. These results underscore JOWA's capacity for rapid and sample-efficient transfer to novel games, highlighting the efficacy of its learned general-purpose representation and decision-making capabilities.

## 5.6 ABLATION STUDY

In this section, we conduct a series of ablation studies to evaluate the impact of key design choices in JOWA, including planning, different training losses, and usage of synthetic data. More experiments can be found in Appendix F. These experiments aim to provide empirical evidence to support our designs and offer valuable insights for future research in this domain.

Table 4: Performance on the 15 Atari games for 3 variants of JOWA evaluated with or without planning reported in terms of the IQM HNS.

|  | JOWA-150M | JOWA-70M | JOWA-40M |
|---|---|---|---|
| without planning | 50.8% | 45.1% | 27.1% |
| with planning | 78.9% | 47.6% | 37.1% |
| improvement (↑) | +28.1% | +2.5% | +10.0% |

Table 5: Results on 7 games for JOWA-150M evaluated with different planning algorithms. See Table 19 for raw scores.

|  | w/o planning | MCTS | Ours |
|---|---|---|---|
| FPS | **10.8** | 0.12 | 1.26 |
| Mean HNS | -2% | 22.1% | **37.8%** |
| IQM HNS | 3% | 13.4% | **23.7%** |

**Effects of planning at decision time.** We evaluate our 3 variants models with or without planning and report the IQM HNS across 15 pretrained games in Table 4. Observe that the addition of planning improves 13.5% performance on average and 28.1% on maximum for JOWA-150M. Even without planning, JOWA still exhibits scaling trends with increasing model size, but at a slower rate.

To further demonstrate the efficiency of our planning algorithm, we compare it with MCTS. Because Muzero-style (Schrittwieser et al., 2020) MCTS requires access to the $V$-function and policy networks $\pi$ while JOWA only estimates the optimal $Q$-function, we use $\max_a Q(s, a)$ as the $V$-value $V(s)$ and use an energy-based policy to compute the action probability, *i.e.*, $\pi(\cdot|s) = \text{softmax}(Q(s, \cdot)/t)$, where $t$ is the temperature. We conduct a grid search on the implementation choices and hyperparameters for MCTS and show the details in Appendix F. We report the frames per second (FPS) and HNS on 7 games, where bare JOWA-150M performs poorly, in Table 5. Observe that ours not only is $10\times$ faster than MCTS but also exceeds MCTS by 71.0% on Mean HNS. Moreover, we empirically observe that MCTS is highly sensitive to the temperature and max depth of the tree while the long execution time makes it inconvenient to tune hyperparameters.

The subsequent ablation experiments require training from scratch in multi-game regime. For time-saving, we consider a subset of 6 games in the following experiments: `Assault`, `Carnival`, `Centipede`, `NameThisGame`, `Phoenix`, `SpaceInvaders`. We train all models with 10% randomly subsampled data per game for 1M gradient steps and fix the parameter size to 40M.

Table 6: The mean, median, and IQM human-normalized score on the 6 Atari games for various training choices. See Tabel 20 in Appendix F for raw scores.

| Game | Origin | Different training losses | | | | Synthetic data in pretraining |
|---|---|---|---|---|---|---|
|  |  | No CQL | No $\mathcal{L}_{\text{world}}$ | sg($\mathcal{L}_{\text{action}}$) | MSE | |
| Mean HNS | **1.183** | 0.613 | 0.917 | 0.293 | 0.189 | 0.448 |
| Median HNS | **1.078** | 0.696 | 0.489 | 0.304 | 0.118 | 0.452 |
| IQM HNS | **1.123** | 0.637 | 0.659 | 0.307 | 0.126 | 0.464 |

**Different training losses.** We train models with each of the following 5 loss functions to investigate the impact of each loss term: (i) original loss defined in Equation (9), (ii) No CQL regularization in $\mathcal{L}_{\text{action}}$, which means no conservative constraints, (iii) No $\mathcal{L}_{\text{world}}$, which means training a model-free transformer-based critic network using $\mathcal{L}_{\text{action}}$, (iv) gradients of $\mathcal{L}_{\text{action}}$ not back-propagated to transformer backbone, denoted as sg($\mathcal{L}_{\text{action}}$) for short, which means optimizing the world model

and critic network separately, (v) mean square error (MSE) instead of distributional TD-loss. The results of different training losses are shown in the third main column of Table 6.

Observe that the $\mathrm{sg}(\mathcal{L}_{\mathrm{action}})$ and MSE loss fail to train qualified multi-game agents. We empirically observe that agents trained with MSE TD-loss always over-optimize the CQL regularization term regardless of the value of coefficient $\alpha$ (tested with $\alpha \in \{0.01, 0.05, 0.1\}$), resulting in extremely high Q-values for in-domain state-actions and low Q-values for OOD actions. The fail of $\mathrm{sg}(\mathcal{L}_{\mathrm{action}})$ underscores the importance of joint-optimization in the world-action model. Comparing the original loss with the no $\mathcal{L}_{\mathrm{world}}$ configuration demonstrates that scaling model-based multi-game RL is more efficient than model-free RL. Additionally, the CQL conservatism regularizer is also necessary in multi-game pretraining. Overall, the original loss outperforms all variant losses by a large margin, indicating the effectiveness of every loss terms.

**Effects of synthetic data in pretraining.** By default, we do not employ planning or synthetic data during pretraining due to the significant increase in training time (approximately $10\times$ slower, extending training to months). For this ablation study, we enable synthetic data in pretraining for JOWA-40M and train on 6 games for 1M steps, which takes 31 days. Specifically, we sample batch of 4-step segments and interact in the imagined MDP with $\epsilon$-greedy policy to synthesis the last 4 steps. Then we optimize JOWA with half real data and half synthetic data using COMBO loss as $\mathcal{L}_{\mathrm{action}}$. The results are shown in the last main column of Table 6.

Suprisingly, we observe negative gains from the usage of synthetic data. We hypothesize twofold reasons: (i) accumulation of inference errors over steps causes later synthetic steps to deviate significantly from the ground-truth distribution, (ii) COMBO's over-penalization of Q-values for unseen state-action pairs results in overly conservative agents. Due to the unbearable training time, further investigation into more effective methods of utilizing synthetic data is left for future work.

## 6   CONCLUSION

In this work, we introduce JOWA: **J**ointly-**O**ptimized **W**orld-**A**ction model, a single offline model-based RL agent capable of playing multiple Atari games. JOWA uses a shared transformer backbone for both the world modeling and the Q-value criticism, enabling joint optimization. We propose a provably efficient and parallelizable planning algorithm to consistently identify the optimal policy during inference. As we hoped, by increasing model parameters, JOWA unlocks scaling trends in performance and exceed prior large-scale offline RL methods in multi-games regime. Furthermore, by training a large-capacity model on a diverse set of games, we show that JOWA can sample-efficiently adapt to novel games, leveraging its generalizable world model for planning. Our ablation studies validate the efficacy of joint optimization and our planning method, while also demonstrating that scaling offline model-based RL is more efficient than offline model-free RL. To facilitate future research, we will release all training and evaluation codes, along with pretrained model checkpoints.

**Limitations.** Due to the computation resources required, we could not experiment on full Atari 2600 games with complete datasets (Agarwal et al., 2020), as this would take 2 months for training. Additionally, while using synthetic data to train agents is common in single-task online model-based RL, we observe it yields negative gains in multi-game RL pretraining. Therefore, how to effectively use synthetic data for multi-task pretraining is an interesting direction for future work. Lastly, although we observe the scaling trend on IQM aggregated scores, the raw scores in some games are against this trend, *i.e.,* increasing model parameters leads to decreasing performance in specific games. We hypothesis that this inconsistence is an inherent problem of offline RL. Unfortunately, to our knowledge, neither increasing model size nor extending training iterations can solve this issue.

## ACKNOWLEDGMENTS

We are grateful to the anonymous reviewers and editors for their insightful suggestions. This work was partially supported by the National Science and Technology Major Project (2022ZD0117102), the National Natural Science Foundation of China under Grants 62271485 and 62303462, Beijing Natural Science Foundation under grant L241016 and L233005, Chongqing Transportation Technology Project (CQJT-CZKJ2024-04), Sichuan Key-Area Research and Development Program 2024YFHZ0011, the Provincial Key Research and Development Program of Zhejiang (project number: 2022C01129), Ningbo International Science and Technology Cooperation Project (2023H020).

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

## A  PROOF OF EQUATION (10)

Given the initial state-action pair $(s_0, a_0)$, the bellman expectation equation (Sutton, 2018) is written as:

$$\begin{aligned}
Q^\pi(s_0, a_0) &= \mathop{\mathbb{E}}_{s_1 \sim P(\cdot|s_0,a_0)} [r(s_0, a_0) + \gamma V^\pi(s_1)] \\
&= \mathop{\mathbb{E}}_{\substack{s_1,\cdots,s_H \sim P \\ a_1,\cdots,a_{H-1} \sim \pi}} \left[ \sum_{t=0}^{H-1} \gamma^t r(s_t, a_t) + \gamma^H V^\pi(s_H) \right]
\end{aligned} \tag{14}$$

According to the definition of optimal Q-value: $Q^*(s_0, a_0) = \max_\pi Q^\pi(s_0, a_0)$, we have:

$$\begin{aligned}
Q^*(s_0, a_0) &= \max_\pi \mathop{\mathbb{E}}_{\substack{s_1,\cdots,s_H \sim P \\ a_1,\cdots,a_{H-1} \sim \pi}} \left[ \sum_{t=0}^{H-1} \gamma^t r(s_t, a_t) + \gamma^H V^\pi(s_H) \right] \\
&= \max_\pi \mathop{\mathbb{E}}_{\substack{s_1,\cdots,s_H \sim P \\ a_1,\cdots,a_{H-1} \sim \pi}} \left[ \sum_{t=0}^{H-1} \gamma^t r(s_t, a_t) + \gamma^H \max_\pi V^\pi(s_H) \right] \\
&= \max_\pi \mathop{\mathbb{E}}_{\substack{s_1,\cdots,s_H \sim P \\ a_1,\cdots,a_{H-1} \sim \pi}} \left[ \sum_{t=0}^{H-1} \gamma^t r(s_t, a_t) + \gamma^H V^*(s_H) \right] \\
&= \max_\pi \mathop{\mathbb{E}}_{\substack{s_1,\cdots,s_H \sim P \\ a_1,\cdots,a_{H-1} \sim \pi}} \left[ \sum_{t=0}^{H-1} \gamma^t r(s_t, a_t) + \gamma^H \max_{a_H} Q^*(s_H, a_H) \right]
\end{aligned} \tag{15}$$

For Equation (15), we derive the optimal policy $\pi$ is the greedy policy selecting actions with the greatest Q*-value as follows:

$$\begin{aligned}
Q^*(s_0, a_0) &= \max_\pi \mathop{\mathbb{E}}_{\substack{s_1,\cdots,s_H \sim P \\ a_1,\cdots,a_{H-1} \sim \pi}} \left[ \sum_{t=0}^{H-1} \gamma^t r(s_t, a_t) + \gamma^H \max_{a_H} Q^*(s_H, a_H) \right] \\
&= \max_\pi \mathop{\mathbb{E}}_{\substack{s_1,\cdots,s_{H-1} \sim P \\ a_1,\cdots,a_{H-2} \sim \pi}} \left[ \sum_{t=0}^{H-2} \gamma^t r(s_t, a_t) + \gamma^{H-1} \max_{a_{H-1}} \mathop{\mathbb{E}}_{s_H \sim P} \left[ r(s_{H-1}, a_{H-1}) + \gamma \max_{a_H} Q^*(s_H, a_H) \right] \right] \\
&= \max_\pi \mathop{\mathbb{E}}_{\substack{s_1,\cdots,s_{H-1} \sim P \\ a_1,\cdots,a_{H-2} \sim \pi}} \left[ \sum_{t=0}^{H-2} \gamma^t r(s_t, a_t) + \gamma^{H-1} \max_{a_{H-1}} Q^*(s_{H-1}, a_{H-1}) \right]
\end{aligned} \tag{16}$$

Therefore, we have $a_{H-1} = \arg\max_a Q^*(s_{H-1}, a)$. Similarly, continuing to use dynamic programming on Equation (16), we finally get: $a_i = \arg\max_a Q^*(s_i, a), (i = 1, 2, \cdots, H-1)$. Thus we claim that the optimal policy is induced by the optimal value and use the symbol $\pi_{Q^*}$ instead of $\pi$ in Equation (15) to obtain Equation (10).

## B  DERIVATION FROM EQUATION (10) TO EQUATION (11)

Given the learned dynamic preditor $\hat{P}$, reward predictor $\hat{r}$, and estimated optimal Q-value $\hat{Q}^*$, we first substitute these three functions for the ground-truth functions in the right side of Equation (10):

$$\max_{\pi_{\hat{Q}^*}} \mathop{\mathbb{E}}_{\substack{s_1,\cdots,s_H \sim \hat{P} \\ a_1,\cdots,a_{H-1} \sim \pi_{\hat{Q}^*}}} \left[ \sum_{t=0}^{H-1} \gamma^t \hat{r}(s_t, a_t) + \gamma^H \max_{a_H} \hat{Q}^*(s_H, a_H) \right] \tag{17}$$

However, due to the estimation error between learned functions and ground-truth functions, $\hat{Q}^*(s_t, a_t)$ is typically not equal to $\mathbb{E}_{s_{t+1} \sim \hat{P}} \left[ \hat{r}(s_t, a_t) + \gamma \max_a \hat{Q}^*(s_{t+1}, a) \right]$. Therefore, instead of using dynamic programming to derive that the optimal policy is the greedy policy as in Proof A, we have to use search to solve formula (17).

However, we can use the conclusion derived with ground-truth functions to make assumptions to reduce the policy space for search. We assume that the optimal policy for formula (17) maps states to actions with top-K highest Q-values. Denote the constrained policy space as $\Pi_{\hat{Q}^*}$, where $\forall \pi \in \Pi_{\hat{Q}^*}, \forall s \in \mathcal{S}, \forall a \notin$ top-$K(\hat{Q}^*(s, \cdot))$, we have $\pi(a|s) = 0$. we make the following assumption: the optimal policy of formula (17) is in the constrained policy space, *i.e.*, $\pi_{\hat{Q}^*}^* \in \Pi_{\hat{Q}^*}$. Under this assumption, formula (17) is equivalent to:

$$\max_{\substack{\pi \in \Pi_{\hat{Q}^*}}} \mathbb{E}_{\substack{s_1, \cdots, s_H \sim \hat{P} \\ a_1, \cdots, a_{H-1} \sim \pi}} \left[ \sum_{t=0}^{H-1} \gamma^t \hat{r}(s_t, a_t) + \gamma^H \max_{a_H} \hat{Q}^*(s_H, a_H) \right] \tag{18}$$

Then we maximize formula (18) over $a_0$ to find the optimal initial action. Considering the above assumption, $a_0 \in$ top-$K(\hat{Q}^*(s_0, \cdot))$. Thus we have the objective for search as:

$$\max_{\substack{\pi \in \Pi_{\hat{Q}^*}}} \mathbb{E}_{\substack{s_1, \cdots, s_H \sim \hat{P} \\ a_0, \cdots, a_{H-1} \sim \pi}} \left[ \sum_{t=0}^{H-1} \gamma^t \hat{r}(s_t, a_t) + \gamma^H \max_{a_H} \hat{Q}^*(s_H, a_H) \right] \tag{19}$$

## C  UPPER BOUND OF SEARCH-BASED Q-VALUE ESTIMATION

Let function $f(s_0, a_0)$ be equal to formula (18) and let $\pi_{\hat{Q}^*}^*$ be the optimal policy of formula (18). We have:

$$f(s_0, a_0) = \mathbb{E}_{\substack{s_1, \cdots, s_H \sim \hat{P} \\ a_1, \cdots, a_{H-1} \sim \pi_{\hat{Q}^*}^*}} \left[ \sum_{t=0}^{H-1} \gamma^t \hat{r}(s_t, a_t) + \gamma^H \max_{a_H} \hat{Q}^*(s_H, a_H) \right] \tag{20}$$

We make the following assumption similar to EfficientZero-v2 (Wang et al., 2024):

**Assumption C.1.** Assume the state transition, reward, and Q-value estimations error are upper bounded by $\epsilon_s, \epsilon_r, \epsilon_Q$ respectively. The error bound of each estimation is formulated as:

$$\max_{n \in [N], t \in [H(n)]} \mathbb{E}\left[ \|\hat{s}_t - s_t\| \right] \leq \epsilon_s \tag{21}$$

$$\max_{n \in [N], t \in [H(n)]} \mathbb{E}\left[ \|\hat{r}(s_t) - r(s_t)\| \right] \leq \epsilon_r \tag{22}$$

$$\max_{n \in [N], t \in [H(n)]} \mathbb{E}\left[ \|\hat{Q}^*(s_t) - Q^*(s_t)\| \right] \leq \epsilon_Q \tag{23}$$

**Theorem C.2.** *Define $s_t, a_t$ to be the states and actions resulting from current policy using ground-truth dynamics $P$ and reward function $r$ and similarly define $s_t', a_t'$ using learned functions $\hat{P}$ and $\hat{r}$. Assume the learned reward function $\hat{r}$ to be $L_r$-Lipschitz and the estimated optimal Q-function $\hat{Q}^*$ to be $L_Q$-Lipschitz. Assume the estimation errors of learned functions are bounded as in Assumption C.1. Then we have the error between search-based Q-value estimation $f(s_0, a_0)$ and ground-truth Q-value $Q^*(s_0, a_0)$ bounded as:*

$$\|f(s_0, a_0) - Q^*(s_0, a_0)\| \leq \frac{1 - \gamma^H}{1 - \gamma} \epsilon_r + \left( \frac{\gamma - \gamma^H}{1 - \gamma} \epsilon_r + \gamma^H \epsilon_Q \right) \epsilon_s + \gamma^H \epsilon_Q \tag{24}$$

*Proof.*

$$\|f(s_0, a_0) - Q^*(s_0, a_0)\|$$

$$= \left\| \mathbb{E}\left[ \sum_{t=0}^{H-1} \gamma^t \hat{r}(s_t', a_t') + \gamma^H \max_{a_H} \hat{Q}^*(s_H', a_H) \right] - \mathbb{E}\left[ \sum_{t=0}^{H-1} \gamma^t r(s_t, a_t) + \gamma^H \max_{a_H} Q^*(s_H, a_H) \right] \right\|$$

$$\leq \mathbb{E}\left[ \left\| \hat{r}(s_0, a_0) - r(s_0, a_0) + \sum_{t=1}^{H-1} \gamma^t \left( \hat{r}(s_t', a_t') - r(s_t, a_t) \right) + \gamma^H \left( \max_{a_H} \hat{Q}^*(s_H', a_H) - \max_{a_H} Q^*(s_H, a_H) \right) \right\| \right]$$

$$\leq \mathbb{E}\left[ \|\hat{r}(s_0, a_0) - r(s_0, a_0)\| \right] + \sum_{t=1}^{H-1} \gamma^t \left\| \hat{r}(s_t', a_t') - r(s_t, a_t) \right\| + \gamma^H \mathbb{E}\left[ \left\| \max_{a_H} \hat{Q}^*(s_H', a_H) - \max_{a_H} Q^*(s_H, a_H) \right\| \right]$$

$$\leq \epsilon_r + \sum_{t=1}^{H-1} \gamma^t \left\| \hat{r}(s_t', a_t') - r(s_t, a_t) \right\| + \gamma^H \mathbb{E}\left[ \left\| \max_{a_H} \hat{Q}^*(s_H', a_H) - \max_{a_H} Q^*(s_H, a_H) \right\| \right] \qquad (25)$$

For the second term in inequality (25):

$$\left\| \hat{r}(s_t', a_t') - r(s_t, a_t) \right\|$$

$$= \left\| \hat{r}(s_t', a_t') - \hat{r}(s_t, a_t) + \hat{r}(s_t, a_t) - r(s_t, a_t) \right\|$$

$$\leq \left\| \hat{r}(s_t', a_t') - \hat{r}(s_t, a_t) \right\| + \|\hat{r}(s_t, a_t) - r(s_t, a_t)\|$$

$$\leq L_r \left\| s_t' - s_t \right\| + \epsilon_r$$

$$\leq L_r \epsilon_s + \epsilon_r \qquad (26)$$

For the third term in inequality (25), let $a_H^1 = \arg\max_{a_H} \hat{Q}^*(s_H', a_H)$ and $a_H^2 = \arg\max_{a_H} Q^*(s_H, a_H)$. Then we have:

$$\mathbb{E}\left[ \left\| \max_{a_H} \hat{Q}^*(s_H', a_H) - \max_{a_H} Q^*(s_H, a_H) \right\| \right]$$

$$= \mathbb{E}\left[ \left\| \hat{Q}^*(s_H', a_H^1) - Q^*(s_H, a_H^2) \right\| \right]$$

$$= \mathbb{E}\left[ \left\| \hat{Q}^*(s_H', a_H^1) - \hat{Q}^*(s_H, a_H^2) + \hat{Q}^*(s_H, a_H^2) - Q^*(s_H, a_H^2) \right\| \right]$$

$$\leq \mathbb{E}\left[ \left\| \hat{Q}^*(s_H', a_H^1) - \hat{Q}^*(s_H, a_H^2) \right\| \right] + \mathbb{E}\left[ \left\| \hat{Q}^*(s_H, a_H^2) - Q^*(s_H, a_H^2) \right\| \right]$$

$$\leq L_Q \left\| s_H' - s_H \right\| + \epsilon_Q$$

$$\leq L_Q \epsilon_s + \epsilon_Q \qquad (27)$$

Substitute inequalities (26) and (27) into (25):

$$\|f(s_0, a_0) - Q^*(s_0, a_0)\|$$

$$\leq \epsilon_r + \sum_{t=1}^{H-1} \gamma^t (L_r \epsilon_s + \epsilon_r) + \gamma^H (L_Q \epsilon_s + \epsilon_Q)$$

$$= \frac{1 - \gamma^H}{1 - \gamma} \epsilon_r + \left( \frac{\gamma - \gamma^H}{1 - \gamma} \epsilon_r + \gamma^H \epsilon_Q \right) \epsilon_s + \gamma^H \epsilon_Q \qquad (28)$$

$\square$

**Analysis.** We expect the search-based Q-values $f(s, a)$ have an upper error bound no greater than the estimated Q-values $\hat{Q}^*(s, a)$, which is formulated as the following inequality:

$$\frac{1 - \gamma^H}{1 - \gamma} \epsilon_r + \left( \frac{\gamma - \gamma^H}{1 - \gamma} \epsilon_r + \gamma^H \epsilon_Q \right) \epsilon_s + \gamma^H \epsilon_Q \leq \epsilon_Q \qquad (29)$$

Based on inequality (29), we derive the following condition:

$$\frac{1}{1-\gamma}\epsilon_r + \left( \frac{\gamma}{1-\gamma} \frac{L_r - \gamma^H L_Q}{1-\gamma^H} - \frac{L_r - L_Q}{1-\gamma} \frac{\gamma^H}{1-\gamma^H} \right) \epsilon_s \leq \epsilon_Q \qquad (30)$$

The inequality (30) means that if the weighted sum of rewards estimation error $\epsilon_r$ and state transition estimation error $\epsilon_s$ are less than or equal to the Q-values estimation error $\epsilon_Q$, then the search-based optimal Q-values have a lower upper-bound of error than estimated optimal Q-values.

## D  GAMES

We select 20 Atari games maintaining the difficulty distribution of full Atari 2600 games defined by Gulcehre et al. (2020), which includes 9 easy games, 9 medium games, and 2 hard games. We use 15 out of 20 games for training and the remaining 5 for OOD generalization experiments. The 15 training games are: `Phoenix`, `Centipede`, `SpaceInvaders`, `Carnival`, `NameThisGame`, `Assault`, `Atlantis`, `DemonAttack`, `BeamRider`, `ChopperCommand`, `Seaquest`, `TimePilot`, `StarGunner`, `Berzerk`, `Zaxxon`. The 5 held-out games are: `Pong`, `Robotank`, `YarsRevenge`, `Gravitar`, `MsPacman`. Details about the size of action spaces and game difficulties are shown in Table 7.

Table 7: Atari Games: Name, Game difficulty, Action Space, and Type.

| Game | Difficulty | Action Space | Type |
|---|---|---|---|
| Assault | Medium | 7 | Train |
| Atlantis | Hard | 18 | Train |
| BeamRider | Medium | 9 | Train |
| Berzerk | Hard | 18 | Train |
| Carnival | Medium | 6 | Train |
| Centipede | Medium | 18 | Train |
| ChopperCommand | Easy | 18 | Train |
| DemonAttack | Easy | 6 | Train |
| Gravitar | Easy | 18 | Fine-tune |
| MsPacman | Medium | 9 | Fine-tune |
| NameThisGame | Easy | 6 | Train |
| Phoenix | Easy | 8 | Train |
| Pong | Medium | 6 | Fine-tune |
| Robotank | Medium | 18 | Fine-tune |
| Seaquest | Easy | 18 | Train |
| SpaceInvaders | Easy | 6 | Train |
| StarGunner | Medium | 18 | Train |
| TimePilot | Easy | 10 | Train |
| YarsRevenge | Medium | 18 | Fine-tune |
| Zaxxon | Easy | 18 | Train |

## E  EXPERIMENTAL DETAILS

### E.1  IMPLEMENT DETAILS

#### E.1.1  JOWA

We implement JOWA based on the codes of IRIS (Micheli et al., 2022)[1]. We train the tokenizer VQ-VAE using the following loss function:

$$\mathcal{L}(E, D, \mathcal{E}) = \|x - D(z)\|_1 + \|\text{sg}(E(x)) - \mathcal{E}(z)\|_2^2 + \|\text{sg}(\mathcal{E}(z)) - E(x)\|_2^2 + \mathcal{L}_{perceptual}(x, D(z))$$

where $E, D, \mathcal{E}$ are encoder, decoder, and embedding table respectively. $\text{sg}(\cdot)$ is the stop-gradient operator. The last term is the perceptual loss (Johnson et al., 2016). We list the hyperparameters of VQ-VAE in Table 8 and 9. After the first stage of pretraining, the VQ-VAE is frozen.

---

[1] https://github.com/eloialonso/iris

Table 8: Encoder / Decoder hyperparameters. We list the hyperparameters for the encoder, the same ones apply for the decoder.

| Hyperparameter | Value |
|---|---|
| Frame dimensions (h, w) | $84 \times 84$ |
| Layers | 3 |
| Residual blocks per layer | 2 |
| Channels in convolutions | 64 |
| Self-attention layers at resolution | 6 / 12 |

Table 9: Embedding table hyperparameters.

| Hyperparameter | Value |
|---|---|
| Vocabulary size | 2048 |
| Tokens per frame (K) | 36 |
| Token embedding dimension | 512 |

In addition to the vocabulary embedding and position embedding, we add a learnable task embedding for observation tokens and action tokens respectively. Our transformer are based on minGPT[2] with FlashAttention (Dao et al., 2022) for acceleration. The hyperparameters of JOWA's transformer backbone are listed in Table 10 and 11.

Table 10: Same hyperparameters of transformer for 3 JOWA variants.

| Hyperparameter | Value |
|---|---|
| max sequence tokens | 296 |
| dropout rate | 0.1 |

Table 11: Different hyperparameters of transformer for 3 JOWA variants.

| Model | Layers | Hidden size | Heads |
|---|---|---|---|
| JOWA-40M | 4 | 512 | 8 |
| JOWA-70M | 6 | 768 | 12 |
| JOWA-150M | 12 | 768 | 12 |

Table 12: Hyperparameters of Q-heads for 3 JOWA variants.

| Model | Layers | MLP Hidden dimension | Number of heads | Dropout |
|---|---|---|---|---|
| JOWA-40M | 3 | 768 | 1 | 0.01 |
| JOWA-70M | 3 | 1024 | 1 | 0.01 |
| JOWA-150M | 3 | 1792 | 3 | 0.01 |

The observation predictor, reward predictor, and terminal predictor are 2-layers MLP. The Q-heads are MLP with dropout, layer normalization, and Mish activations from Hansen et al. (2024). The hyperparameters of Q-heads for JOWA are shown in Table 12. The training hyperparameters of JOWA are shown in Table 13.

### E.1.2 MTBC

We implement MTBC based on JOWA. We remove the observation predictor, reward predictor, and terminal predictor. We change the output dimension of Q-heads to 18 and train the heads as a 18-class classification problem. All hyperparameters are kept the same as JOWA.

### E.1.3 EDT

We use the official code for EDT[3]. We implement EDT-200M based on the architecture configuration of MGDT-200M, which is shown in Table 14. We change the batch size to 512 and keep other hyperparameters the same as its original configuration. We enable data augmentation (random cropping and random rotation) for EDT.

### E.1.4 MGDT

We implement MGDT based on the codes of EDT. We use $\{o_{t+i}, R_{t+i}, a_{t+i}, r_{t+i}\}_{i=0}^{3}$ as the input sequences, remove the expectile regression loss $\mathcal{L}_{\max}$ and observation prediction loss $\mathcal{L}_{\text{observation}}$, and

---

[2]https://github.com/karpathy/minGPT
[3]https://github.com/kristery/Elastic-DT

Table 13: Training hyperparameters of JOWA.

| Hyperparameter | Value |
|---|---|
| Optimizer (VQ-VAE) | Adam |
| Optimizer (except VQ-VAE) | AdamW |
| Learning rate (VQ-VAE) | 0.0001 |
| Learning rate (except VQ-VAE, stage 1) | 0.0001 |
| Learning rate (except VQ-VAE, stage 2) | 0.00005 |
| Batch size (VQ-VAE) | 2048 |
| Batch size (except VQ-VAE) | 512 |
| Weight decay (except VQ-VAE) | 0.01 |
| Gradient clip | 1.0 |
| Discount factor ($\gamma$) | 0.99 |
| Target Q update frequency | 1000 |
| Distributional Q | [-10, 30] |
| Number of atoms | 51 |
| Coefficient of CQL ($\alpha$) | 0.1 |
| Coefficient of $\mathcal{L}_{\text{world}}$ ($\beta$) | 0.1 |

Table 14: Hyperparameters of transformer for 2 MGDT variants.

| Model | Layers | Hidden size | Heads |
|---|---|---|---|
| MGDT-40M | 6 | 768 | 12 |
| MGDT-200M | 10 | 1280 | 20 |

add the reward prediction loss to rewrite the codes of EDT into MGDT. We enable data augmentation (random cropping and random rotation) for MGDT. The hyperparameters of transformer for two MGDT variants are listed in Table 14.

### E.1.5 SCALED-QL

We implement a pytorch version of Scaled-QL from scratch, referring to the jax version of its official preliminary codes[4]. We use ResNet-101 as the representation backbone, followed by 3-layers MLP with 1024 hidden neurons and an output layer. We replace the batch normalization in ResNet with group normalization and use a learnable spatial embeddings to aggregate the outputs of the ResNet instead of global mean pooling. Before the output layer, we normalize the feature $e$ as $\frac{e}{\|e\|^2}$. The training hyperparameters of Scaled-QL are listed in Table 15.

For fair comparison, all methods are trained with the same batch size of 512 for 1.75M gradient steps. For reporting results, we report the performance of the agent at the end of pretraining.

### E.2 FINE-TUNING PROTOCOL

We uniformly draw 5k transitions from expert-level DQN-Replay (Agarwal et al., 2020) (last 20% of the original dataset) for each held-out game. Each game was fine-tuned separately to measure the model's transfer performance for a fixed game. we fine-tuned all methods using a batch size of 32 and learning rate of 0.00005 for 50k gradient steps. For reporting results, we report the performance of the agent snapshot that obtain the highest score during fine-tuning.

For JOWA in the second fine-tuning stage, we set both the planning horizon and the beam width to 2 for all fine-tuning experiments. Thus we sample batch of 6-steps segments, using planning algorithm to synthesis the last 2 steps. For other baselines, we enable random cropping and random rotation for data augmentation.

---

[4]https://tinyurl.com/scaled-ql-code

Table 15: Training hyperparameters of Scaled-QL.

| Hyperparameter | Value |
|---|---|
| Optimizer | Adam |
| Learning rate | 0.0002 |
| Batch size | 512 |
| Gradient clip | 1.0 |
| Discount factor ($\gamma$) | 0.99 |
| Target Q update frequency | 2000 |
| Distributional Q | [-20, 20] |
| Number of atoms | 51 |
| Coefficient of CQL ($\alpha$) | 0.05 |
| n-step returns | 3 |

Table 16: Evaluation settings of Atari.

| Hyperparameter | Value |
|---|---|
| Sticky actions | No |
| Grey-scaling | True |
| Observation down-sampling | (84, 84) |
| Frames stacked | 4 |
| Frame skip (Action repetitions) | 4 |
| Terminal condition | Game Over |
| Max frames per episode | 108K |
| Evaluation noise $\epsilon_{\text{eval}}$ | 0.001 |

### E.3 EVALUATION PROTOCOL

For all methods, each game score is calculated by averaging over 16 model rollout episode trials. To reduce inter-trial variability, we do not use sticky actions during evaluation following Lee et al. (2022); Kumar et al. (2023). Following standard protocols on Atari, we evaluate a noised version of the policy with an epsilon-greedy scheme, with $\epsilon_{\text{eval}} = 0.001$. The evaluation settings of Atari are shown in Table 16.

For the expert action inference of MGDT and EDT, we set the inverse temperature $\kappa$ to 10. For the planning of JOWA, we set the planning horizon $H$ to 2 for all games. The beam width are set according to the size of valid action space of each game. Specifically, we set beam width $K$ to 2 if the valid action space size is less than 10, otherwise we set $K$ in $\{3, 4\}$. The planning hyperparameters for each game are shown in Table 17.

Table 17: Planning hyperparameters of JOWA during evaluation.

| Game | planning horizon | beam width | Action space |
|---|---|---|---|
| Assault | 2 | 2 | 7 |
| Atlantis | 2 | 3 | 18 |
| BeamRider | 2 | 2 | 9 |
| Berzerk | 2 | 4 | 18 |
| Carnival | 2 | 2 | 6 |
| Centipede | 2 | 4 | 18 |
| ChopperCommand | 2 | 4 | 18 |
| DemonAttack | 2 | 2 | 6 |
| NameThisGame | 2 | 2 | 6 |
| Phoenix | 2 | 2 | 8 |
| Seaquest | 2 | 3 | 18 |
| SpaceInvaders | 2 | 2 | 6 |
| StarGunner | 2 | 3 | 18 |
| TimePilot | 2 | 3 | 10 |
| Zaxxon | 2 | 3 | 18 |

## F MORE EXPERIMENTS AND RAW SCORES

We summarize the raw scores of fine-tuning experiments, ablation studies on planning algorithms and training choices in Table 18, 19 and 20 respectively.

**Details of the comparison of planning algorithms.** We implement both planning algorithms in python rather than C++ for fair speed comparison. Note that Muzero-style (Schrittwieser et al., 2020) MCTS requires access to the $V$-function and policy networks $\pi$ while JOWA only estimates the optimal $Q$-function. Therefore, we conduct a grid search on the following choices for MCTS:

1. Compute the $V$-value using `Q.mean()` or `Q.max()`.

Table 18: Offline fine-tuning performance on unseen games using 5k transitions, measured in terms of DQN-normalized score, following Lee et al. (2022); Kumar et al. (2023).

| Game | Random | DQN | MTBC 120M | MGDT 200M | EDT 200M | SQL 80M | JOWA 150M | JOWA-150M (scratch) |
|---|---|---|---|---|---|---|---|---|
| Gravitar | 173.0 | 473.0 | 35.7 | 250.0 | 253.3 | 137.5 | 273.3 | 83.3 |
| MsPacman | 307.3 | 3085.6 | 905.0 | 1290.3 | 1210.7 | 1040.2 | 2016.7 | 786.7 |
| Pong | -20.7 | 19.5 | 5.8 | 9.7 | 11.3 | 13.7 | 17.7 | 8.8 |
| Robotank | 2.2 | 63.9 | 6.8 | 16.0 | 15.5 | 19.7 | 25.0 | 11.0 |
| YarsRevenge | 3092.9 | 18089.9 | 7987.5 | 10886.3 | 11276.9 | 10838.5 | 17506.2 | 6507.0 |
| Mean | 0.000 | 1.000 | 0.164 | 0.422 | 0.430 | 0.360 | **0.647** | 0.196 |
| Median | 0.000 | 1.000 | 0.215 | 0.354 | 0.325 | 0.284 | **0.615** | 0.173 |
| IQM | 0.000 | 1.000 | 0.205 | 0.377 | 0.380 | 0.355 | **0.647** | 0.181 |

Table 19: Raw scores on 7 games for JOWA-150M evaluated with different planning algorithms.

| Game | Random | Human | w/o planning | MCTS | Ours |
|---|---|---|---|---|---|
| BeamRider | 363.9 | 16926.5 | 864.5 | 1137.0 | **3498** |
| Berzerk | 123.7 | 2630.4 | 396.9 | 440.0 | **739** |
| Carnival | 0 | 3800 | **5560.0** | 3340.0 | 5316 |
| ChopperCommand | 811.0 | 7387.8 | 806.2 | 1850.0 | **3812.5** |
| Seaquest | 68.4 | 42054.7 | 267.5 | 760.0 | **2725** |
| TimePilot | 3568 | 5229.2 | 662.5 | **4100.0** | 3669 |
| Zaxxon | 32.5 | 9173.3 | 12.5 | 50.0 | **2163** |
| #Mean HNS | 0.000 | 1.000 | -0.02 | 0.221 | **0.378** |
| IQM HNS | 0.000 | 1.000 | 0.03 | 0.134 | **0.237** |

2. We employ an energy-based policy to compute action probability, *i.e.*, $\pi(\cdot|s) = \text{softmax}(Q(s,\cdot)/t)$, where $t$ is the temperature. We search $t$ in $\{0.01, 0.1, 0.5, 0.7, 0.8, 0.9, 1, 2, 3, 5, 10\}$.

3. Use the action of most visited or most valuable children of the root state as the optimal action.

4. Search the max depth of the tree $H$ in $[1, 7]$.

Finally, we use the configuration with `V=Q.max()`, $t = 0.9$, and action of most valuable children as the optimal action for all games while searching $H$ in $\{1, 2, 4, 6\}$ for each game. Even though we believe that we conduct a sufficiently adequate hyperparameter search, MCTS still performs worse than ours, as shown in Table 19.

We compare these two algorithms on 7 games where bare JOWA-150M (*i.e.*, without planning) performs poorly. The results show that our planning algorithm exceeds MCTS by 71.0% Mean HNS. Moreover, we find that MCTS is highly sensitive to the temperature $t$ and max depth $H$, and inappropriate values of hyperparameters can even degenerate the policy into a randomized policy. However, its long execution time makes it inconvenient to tune hyperparameters. Because of this, we have not yet found hyperparameters that can make MCTS perform reasonably on the other 8 pretrained games. We will include MCTS as an optional planning algorithm in our open-source evaluation code.

**Ensemble of Q-value heads.** We run JOWA-40M with ensembled Q-heads, where the return distribution is a weighted sum of distributions from multiple Q-heads. The original configuration of JOWA-40M uses a single Q-head. In this experiment, we set the number of Q-heads to 3 and compare 2 ensemble approaches: equal weights or random weights like REM (Agarwal et al., 2020). We report the results in the forth main column of Table 20. Although the equal weighted ensemble slightly outperforms random weights, we have not observed significant improvements of multi Q-heads over single Q-head. We leave the better way of distributional-Q ensemble for future work.

**Effects of task embedding.** We run JOWA-40M without task embedding, using only the sum of vocabulary embedding and position embedding as input for transformer. We report results in the

Table 20: Raw scores on the 6 Atari games for various training choices. The mean, median, and IQM human-normalized score are shown in the last 3 rows and the best scores are markded in bold.

| Game | Origin | Different training losses | | | | Q-heads ensemble | | No task embedding | Synthetic data in pretraining |
|---|---|---|---|---|---|---|---|---|---|
| | | No CQL | No $\mathcal{L}_{world}$ | sg($\mathcal{L}_{action}$) | MSE | Equal | Random | | |
| Assault | 1423.5 | 857.9 | 1650 | 452.7 | 637.6 | 1628.8 | 1258.6 | 1428.5 | 655.7 |
| Carnival | 5560 | 4144.6 | 5560 | 2120 | 620 | 5154.3 | 4160 | 5974.2 | 3492.9 |
| Centipede | 5018.9 | 1494.1 | 8146 | 5568.7 | 4097.3 | 5592.5 | 6450 | 3725.4 | 3253 |
| NameThisGame | 12208.1 | 9307.1 | 4407.3 | 2108.8 | 1315 | 12148.6 | 9420 | 7700 | 5080 |
| Phoenix | 4740 | 140 | 2036.7 | 1920 | 1190 | 4610 | 4941.7 | 4020 | 193.3 |
| SpaceInvaders | 1201.7 | 605 | 323.4 | 539.3 | 260 | 958.4 | 1283.3 | 575.4 | 786.3 |
| Mean HNS | 1.183 | 0.613 | 0.917 | 0.293 | 0.189 | **1.209** | 1.026 | 0.964 | 0.448 |
| Median HNS | **1.078** | 0.696 | 0.489 | 0.304 | 0.118 | 0.975 | 0.921 | 0.721 | 0.452 |
| IQM HNS | **1.123** | 0.637 | 0.659 | 0.307 | 0.126 | 1.049 | 0.931 | 0.824 | 0.464 |

Table 21: Offline fine-tuning performance of JOWA-150M on unseen games with various data quality, measured in terms of DQN-normalized score, following Lee et al. (2022); Kumar et al. (2023).

| Game | Random | DQN | Expert | Suboptimal | Highly-suboptimal |
|---|---|---|---|---|---|
| Gravitar | 173.0 | 473.0 | 273.3 | 317.8 | 296.0 |
| MsPacman | 307.3 | 3085.6 | 2016.7 | 1005.5 | 1126.8 |
| Pong | -20.7 | 19.5 | 17.7 | 13.2 | 13.8 |
| Robotank | 2.2 | 63.9 | 25.0 | 14.6 | 8.5 |
| YarsRevenge | 3092.9 | 18089.9 | 17506.2 | 15085.0 | 9755.4 |
| Mean | 0.000 | 1.000 | 0.647 | 0.516 | 0.422 |
| Median | 0.000 | 1.000 | 0.615 | 0.483 | 0.410 |
| IQM | 0.000 | 1.000 | 0.647 | 0.511 | 0.383 |

fifth main column of Table 20. Observe that the addition of task embedding improves 21.9% and 35.7% for mean and median human-normalized score respectively.

**Fine-tuning with non-expert data.** Comparing with fine-tuning using expert-level data, we additionally fine-tune JOWA-150M using 5k suboptimal and highly-suboptimal transitions. Specifically, the suboptimal and the highly-suboptimal transitions are uniformly sampled from the complete and the initial 20% of the DQN-Replay dataset, respectively. The results in Table 21 show that the fine-tuning performance strongly correlates with data quality. The mean DQN-normalized scores for expert, suboptimal, and highly-suboptimal data are 0.647, 0.516, and 0.422 respectively.

**Discussion of emergent behaviors.** Capability emergence refers to the phenomenon where large models suddenly exhibit new abilities or significantly enhanced performance, typically occurring after reaching certain scale thresholds in model size. For example, JOWA-150M achieves a significantly higher score than JOWA-70M and other baselines on Zaxxon, which seems like the "emergent" phenomenon. However, after examining the reward of Zaxxon, we conclude that this "emergent behavior" is actually attributable to the nonlinear reward function. In addition to the common rewards of around 100, Zaxxon also defines a huge reward of nearly 5000. JOWA-70M gets this huge reward in 1 of the 16 rollouts while JOWA-150M gets it in 7 rollouts during evaluation. When we clip the rewards to the range $[-1, 1]$, JOWA-70M and JOWA-150M achieve scores of 0.08 and 0.44 respectively, indicating no true "emergent" phenomenon. Schaeffer et al. (2024) draws a similar conclusion that "nonlinear or discontinuous metrics cause the emergent abilities".

