# OpenReview forum: "Scaling Offline Model-Based RL via Jointly-Optimized World-Action Model Pretraining"
_ICLR.cc/2025/Conference — ICLR 2025 Poster_

### Official Review · Reviewer_XU9g · 2024-10-19

**Soundness:** 4
**Presentation:** 3
**Contribution:** 3
**Rating:** 8
**Confidence:** 5

**Summary:**

The paper focuses on the important research question: "Can image observation-based world model scale offline RL across multiple tasks while enhancing generalization to diverse unseen tasks?". The paper presents JOWA, which 1) jointly trains a Transformer model that includes world model, policy, critic using the offline data, 2) planning with beam-search in inference-time using the trained world model. Experimental results in Atari tasks shows that JOWA trained with 10% of subsampled 15 tasks significantly outperforms previous multi-task offline RL approaches not only in in-distribution tasks, but also in fine-tuning with 5k expert demonstrations. Moreover, JOWA shows favorable scaling property respect to the number of parameters. Exhaustive ablation studies show the importance of individal components of JOWA.

**Strengths:**

S1. JOWA significantly outperforms previous multi-task offline RL approaches.
* Experimental results of Table 2 shows that JOWA gives state-of-the art result in multi-task offline RL with 10% subsampled Atari (Mean 0.456, IQM 0.789)
* Experimental results of Table 3 shows that JOWA significantly outperforms prior works in fine-tuning (Mean 0.647, IQM 0.647)

S2. JOWA shows favorable scaling property.
* Increase in the number of parameters leads to better performance.
* Compared to other methods, the performance increase is significant.

S3. Ablation studies shows useful insights on JOWA's performance.
* Planning with world model largely contributes the performance.
* Training policy jointly with the world model is important for multi-task agents.

S4. Open-source codes & checkpoints.
* Those resources will be helpful for the community.

**Weaknesses:**

W1. Evaluated only in Atari
* For example, continous tasks like DMControl are relatively simple but challenging.
* It will be exciting to see if JOWA can be also applied in continuous tasks.

W2. Discrepancy between Theorem 1 and planning of JOWA
* Although the planning emprically improves the performance, it seems that the theory slightly differs with the planning of JOWA. Specifcally, Theorem 1 assumes critic is the estimate of the optimal Q-function, but JOWA uses a conservative estimate (CQL). Thus, $Q(s, a) - Q*(s, a)$ will not be bounded by the term mentioned in Theorem 1; Q(s, a) will be much smaller than Q*(s, a) in reality.
* Please correct me in the point if I'm wrong.

**Questions:**

Q1. Can Theorem 1 be applied for planning of JOWA?
* Theorem 1 shows that error caused by model-based planning can be bounded. However, it assumes that the critic $Q(s,a)$ is the estimate of the optimal Q-function (i.e. $\mathbb{E}[\sum_t \gamma^t r_t]$). However, the critic is learned via CQL, which tends to underestimate the Q-values, as discussed in [1]. Thus, $Q(s, a) - Q*(s, a)$ will not be bounded by the term mentioned in Theorem 1; Q(s, a) will be much smaller than $Q^{*}(s, a) = \mathbb{E}[\sum_t \gamma^t r_t]$ in reality.
* Can you share your thoughts on this? Please correct me in the point if I'm wrong.

Q2. Scaling property in fine-tuning tasks
* Can you provide the fine-tuning results for JOWA-40M, JOWA-70M?
* I think it will be helpful on investigating scaling effects in fine-tuning.

Q3. Extending JOWA to continuous environments
* It would be exciting if JOWA can be also extended to continuous tasks. For example, continous tasks like DMControl are relatively simple but challenging.
* However, I understand that the training process is time-consuming and authors should put a lot of effort on this. Thus, I'm not strongly suggesting (but welcome) the authors to answer this question via additional experiments; if then, telling your thoughts or ideas on how to extend this work to continuous environments will be helpful for potential users of JOWA and future works.

[1] Mitsuhiko Nakamoto et al., Cal-QL: Calibrated Offline RL Pre-Training for Efficient Online Fine-Tuning, NeurIPS 2023.

---

> ### Author Response · Authors · 2024-11-21
> **Rebuttal by Authors [1/2]**
>
> Dear Reviewer XU9g,
>
> We sincerely appreciate your valuable and insightful comments as they are extremely helpful for improving our manuscript. We have addressed each comment in detail in the paragraphs below.
>
> ---
>
> > **[w1]** Evaluated only in Atari
>
> **[A1]** We appreciate your interset in extending JOWA to continous control tasks. We believe that extending JOWA to the continuous control tasks is an interesting and promising direction for future work. We have already started arranging computing resources for this project. Our ultimate goal is to use TD to learn a generalist/multi-task policy (on common benchmarks, such as Atari, ProcGen, DMLab, DMControl, and Meta-World) from the offline dataset while also focusing on efficient adaptation to OOD tasks.
>
> ---
>
> >  **[w2]** Discrepancy between Theorem 1 and planning of JOWA
>
> **[A2]** Thank you for bringing this to our attention. We give the detailed answer in **[A3]**.
>
> ---
>
> > **[Q1]** Can Theorem 1 be applied for planning of JOWA?
>
> **[A3]** Thank you for bringing this to our attention. The problem you mentioned about CQL causing the Q-value underestimation does exist. However, after carefully reviewing the proof of Theorem 1 in Appendix C, we found that the theorem still holds even with this underestimation issue, provided that the error between the CQL-optimized Q-function and the ground-truth optimal Q-function $Q^*$ is bounded by $\epsilon_Q$.
>
> This bounded error assumption necessarily holds for JOWA because we restrict the Q-value to the range [$Q_\min$, $Q_\max$] as required by the C51 algorithm. Therefore, by setting $\epsilon_Q = Q_\max-Q_\min$, the assumption holds, and then the theorem holds.
>
> ---
>
> > **[Q2]** Scaling property in fine-tuning tasks
>
> **[A4]** Thank you for highlighting this. We fine-tuned JOWA-40M and JOWA-70M using the same 5k transitions and present the results in the following table. The results show that the Mean and IQM DQN-normalized scores scales with model size while Median DNS doesn't.
>
> Table 1: Fine-tuning results for JOWA.
>
> | Game        | JOWA-150M | JOWA-70M | JOWA-40M |
> | ----------- | --------- | -------- | -------- |
> | Gravitar    | 273.3     | 387.4    | 326.5    |
> | MsPacman    | 2016.7    | 908.0    | 1296.3   |
> | Pong        | 17.7      | 14.6     | 14.2     |
> | Robotank    | 25.0      | 13.8     | 5.4      |
> | YarsRevenge | 17506.2   | 16339.2  | 14072.1  |
> | #Mean DNS   | 0.647     | 0.576    | 0.504    |
> | Median DNS  | 0.615     | 0.715    | 0.512    |
> | IQM DNS     | 0.647     | 0.603    | 0.498    |

---

> ### Author Response · Authors · 2024-11-21
> **Rebuttal by Authors [2/2]**
>
> > **[Q3]** Extending JOWA to continuous environments.
>
> **[A5]** We appreciate your interset in extending JOWA to continous control tasks. We believe that extending JOWA to the continuous control tasks is an interesting and promising direction for future work. We have already started arranging computing resources for this project. Our ultimate goal is to use TD to learn a generalist/multi-task policy (on common benchmarks, such as Atari, ProcGen, DMLab, DMControl, and Meta-World) from the offline dataset while also focusing on efficient adaptation to OOD tasks.
>
> We sincerely apologize that due to the time limit of the rebuttal period, we have allocated our resources to other experiments, making it impossible to demonstrate JOWA's performance on continuous tasks at present. However, we can outline 3 potential approaches for extending JOWA to continuous environments:
>
> 1. C51-style approach: We divide the value range of each action dimension evenly into $K$ intervals and use the mean of the interval to represent the value of this interval. We then form the discrete action space by taking the union of all possible discrete values and treat it as a discrete control task using C51-style method to train an **action-level optimal Q-function**. While this approach requires minimal code changes, we have concerns about the performance impact of discretization and the exponential growth of the action space in high-dimensional environments.
> 2. Q-Transformer-style approach: After discretizing each action dimension, we learn a **dimension-level (rather than action-level) optimal Q-function** like Q-Transformer[1]. This approach requires moderate code modifications. However, we are concerned that the additional Bellman updates across action dimensions might exacerbate the credit assignment problem in high-dimensional action tasks.
> 3. REINFORCE-style approach: Following action dimension discretization, we learn an action-level Q-function (not optimal Q-function) where the Q-head takes the last action dimension token's embedding as input. We calculate the target Q-value using the Monte Carlo method with the corresponding offline trajectory and treat Q-head training as a multi-class classification problem using CELoss (with CQL penalty) for stability. Then A dimension-level policy head uses each one of embeddings from last observation token to the penultimate action dimension token to predict next action dimension, trained via REINFORCE rather than auto-regression. This framework (**action-level Q-function with dimension-level policy**) seems somehow like ArCHer[2], but we use the shared backbone for Q-network and policy network and also incorporates world model for planning. This method requires substantial code changes and potential re-experimentation on Atari, but in my experience it may be the most stable way for training.
>
> Currently, we have begun preparing computing resources for experiments on small-scale multi-continuous tasks. We plan to start with the second method as we progress toward our ultimate goal.
>
> [1] Chebotar, Yevgen, et al. "Q-transformer: Scalable offline reinforcement learning via autoregressive q-functions." *Conference on Robot Learning*. PMLR, 2023.
>
> [2] Zhou, Yifei, et al. "Archer: Training language model agents via hierarchical multi-turn rl." *arXiv preprint arXiv:2402.19446* (2024).

---

> > ### Comment · Reviewer_XU9g · 2024-11-23
> >
> > Thank you for the authors for the detailed response. Regarding the responses to my main questions:
> >
> > **A1, A5.** I understand that it is challenging to do additional experiments for continuous environments. I believe that the potential approaches shared by the authors will be helpful for future works from JOWA.
> >
> > **A2, A3.** Thank you for reviewing my concern. Now I clearly see that Theorem 1 actually shines when Q estimation is inaccurate, since the model prediction will be much accurate.
> >
> > **A4.** Thank you for providing the experimental result. The result clearly shows the scaling property with IQM and Mean scores on fine-tuning. I think adding this result in the paper will be helpful on showing the scaling property in fine-tuning.
> >
> > Overall, I believe this paper provides a meaningful contribution the offline RL community. I will maintain my score of 8 but have increased my confidence to 5 that it should be accepted.

---

> > > ### Author Response · Authors · 2024-11-23
> > >
> > > Thank you for recognizing our work and for your time and effort as a reviewer! We will add experiments on the scaling property of the fine-tuning task in Sections 5.4 and 5.5 of our main paper.

---

### Official Review · Reviewer_pLGR · 2024-10-30

**Soundness:** 4
**Presentation:** 4
**Contribution:** 3
**Rating:** 6
**Confidence:** 4

**Summary:**

The paper introduces JOWA (Jointly-Optimized World-Action model), a novel offline model-based reinforcement learning (RL) agent designed for generalization across multiple tasks, specifically focusing on scaling RL with image-based observations. JOWA is pretrained on multiple Atari games using a shared transformer backbone that jointly optimizes a world model and action model. This setup allows the model to learn both general-purpose representations and decision-making skills.

JOWA stabilizes temporal difference learning by incorporating world modeling as a regularizer, enabling large models to scale effectively. The proposed framework also features a parallelizable planning algorithm to improve Q-value estimation and policy search at inference time. Experiments show that JOWA outperforms state-of-the-art offline RL methods, achieving 78.9% human-level performance on pretrained games and demonstrating strong sample-efficient transfer to novel games with minimal fine-tuning.

**Strengths:**

- Extension from Sequential Modeling: JOWA effectively adapts transformer-based architectures from NLP to reinforcement learning, leveraging tokenization for image observations and combining offline RL with online planning to scale across tasks.
- Strong Baseline Performance: It outperforms state-of-the-art methods, showing significant improvements on Atari benchmarks while requiring only 10% of the dataset.
- Adaptability: JOWA demonstrates efficient transfer to new tasks with minimal fine-tuning, highlighting its robustness and generalization capabilities.

**Weaknesses:**

- Limited Originality: While JOWA effectively combines existing state-of-the-art techniques, it lacks substantial novelty in introducing new ideas. The core contributions largely build upon well-established methods, such as transformers, tokenization, and offline RL.
- Reliance on Expert Data for New Games: While JOWA demonstrates strong performance in fine-tuning to new tasks, it heavily relies on expert-level data for unseen games. This dependence on high-quality data might limit its applicability in scenarios where such data is not readily available or costly to obtain.
- Excessive Discretization: The paper discretizes the reward into the set {-1, 0, 1}, which may lead to a loss of important nuance and granularity in reward signals. This overly simplified reward structure might not capture the full complexity of more sophisticated environments, limiting the model’s performance in tasks requiring finer reward distinctions.

**Questions:**

- Why was the sequence length relative low (8)?
- Could JOWA learn useful policy on new games using non-expert transitions?
- It seems the number for MGDT are very different from the original paper (MGDT achieved 93% IQM HNS on 40M model). Why is this?
- Is the action-part module (CQL) hard to tune in this method?

---

> ### Author Response · Authors · 2024-11-21
> **Rebuttal by Authors [1/3]**
>
> Dear Reviewer pLGR,
>
> We sincerely appreciate your valuable and insightful comments as they are extremely helpful for improving our manuscript. We have addressed each comment in detail in the paragraphs below.
>
> ---
>
> > **[w1]** Limited Originality.
>
> **[A1]** Thank you for pointing this out. We think that the novelty of JOWA can be examined in the following ways:
>
> 1. JOWA examines which existing technologies are crucial for scaling RL, providing a novel perspective on evaluating established techniques.
> 2. JOWA introduces a novel planning algorithm. Our ablation experiments (Table 4) demonstrate that this planning algorithm significantly improves performance. As detailed in our response [A5] to reviewer zGRd, when compared to MCTS, our algorithm is not only 10$\times$ faster but also achieves better performance.
>
> ---
>
> > **[w2]** Reliance on Expert Data for New Games.
>
> **[A2]** Thank you for your greatful suggestion. We have conducted additional fine-tuning experiments using non-expert transitions. See **[A5]** for details.
>
> ---
>
> > **[w3]** Excessive Discretization of rewards.
>
> **[A3]** Thank you for highlighting this. We want to clarify that using the sign function to clip reward is a common trick in Atari domain [1~4] and is adopted by the Stable Baseline3's Atari wrapper [5]. This trick serves several important purposes:
>
> 1. It helps unify the value range across different games, enabling us to use the same value network architecture for multiple games.
> 2. It reduces the scale disparity between different games, making multi-task learning more stable.
>
> While we acknowledge that this discretization might lose some reward granularity, empirical results from multiple studies have shown its effectiveness in practice, particularly in the Atari domain.
>
> [1] Kaiser, Lukasz, et al. "Model-based reinforcement learning for atari." *arXiv preprint arXiv:1903.00374* (2019).
>
> [2] Micheli, Vincent, Eloi Alonso, and François Fleuret. "Transformers are Sample-Efficient World Models." *The Eleventh International Conference on Learning Representations*.
>
> [3] Kumar, Aviral, et al. "Offline q-learning on diverse multi-task data both scales and generalizes." *arXiv preprint arXiv:2211.15144* (2022).
>
> [4] Lee, Kuang-Huei, et al. "Multi-game decision transformers." *Advances in Neural Information Processing Systems* 35 (2022): 27921-27936.
>
> [5] [Atari Wrappers — Stable Baselines3 0.11.1 documentation](https://stable-baselines3.readthedocs.io/en/v0.11.1/common/atari_wrappers.html)
>
> ---
>
> > **[Q1]** Why was the sequence length relative low (8)?
>
> **[A4]** Thank you for raising this question about the sequence length. We set the sequence length to 8 based on the following considerations:
>
> 1. Computational Efficiency: Given the substantial size of the dataset (~3TB), we can't load it entirely into memory and must read transitions from disk when the dataloader calls. The sequence length directly impacts how many observations we need to read per time. A longer sequence length (e.g., 20) would make data loading twice as slow as the training time (forward+backward), creating a significant bottleneck in training speed.
> 2. Minimum Required Context: Based on DQN experience, we estimate that 4-step observations are sufficient for decision-making. Given our planning algorithm's horizon $H=2$, the minimum required sequence length is 6. We selected 8 as it is the smallest power of 2 exceeding this minimum requirement.
> 3. Prior Research: We also refer to this paper's[1] experimental results on context length (Figure 6 in its appendix) to some extent, although their experiments were conducted on Decision Transformer.
>
> [1] Bhargava, Prajjwal, et al. "When should we prefer Decision Transformers for Offline Reinforcement Learning?." *arXiv preprint arXiv:2305.14550* (2023).

---

> ### Author Response · Authors · 2024-11-21
> **Rebuttal by Authors [2/3]**
>
> > **[Q2]** Could JOWA learn useful policy on new games using non-expert transitions?
>
> **[A5]** We appreciate your interest in fine-tuning JOWA with non-expert transitions. To address this question, we conducted additional fine-tuning experiments using 5k suboptimal and highly-suboptimal transitions. Specifically, the suboptimal transitions were uniformly sampled from the complete DQN-Replay dataset, and the highly-suboptimal transitions were uniformly sampled from the initial 20% of the DQN-Replay dataset.
>
> The results shown in the following table demonstrate that the fine-tuning performance strongly correlates with data quality. The mean DQN-normalized scores for expert, suboptimal, and highly-suboptimal data are 0.647, 0.516, and 0.422 respectively.
>
> Table 1: Fine-tuning experiments of JOWA-150M with various type of transitions.
>
> | Game        | Expert  | Suboptimal | Highly-suboptimal |
> | ----------- | ------- | ---------- | ----------------- |
> | Gravitar    | 273.3   | 317.8      | 296.0             |
> | MsPacman    | 2016.7  | 1005.5     | 1126.8            |
> | Pong        | 17.7    | 13.2       | 13.8              |
> | Robotank    | 25.0    | 14.6       | 8.5               |
> | YarsRevenge | 17506.2 | 15085.0    | 9755.4            |
> | #Mean DNS   | 0.647   | 0.516      | 0.422             |
> | Median DNS  | 0.615   | 0.483      | 0.410             |
> | IQM DNS     | 0.647   | 0.511      | 0.383             |

---

> ### Author Response · Authors · 2024-11-21
> **Rebuttal by Authors [3/3]**
>
> > **[Q3]** It seems the number for MGDT are very different from the original paper (MGDT achieved 93% IQM HNS on 40M model). Why is this?
>
> **[A6]** Thank you for pointing this out. We carefully examined the results from the original MGDT and ours, and found that different game sets lead to substantially different IQM scores even with similar raw scores. For clarity, we present the raw scores of MGDT-40M and -200M from our experiments and the original MGDT paper for our 12 pretrained games (note that other 3 games - Berzerk, SpaceInvaders, and StarGunner - were not pretrained in the original MGDT). Specifically, the MGDT-40M raw scores are from the Scaled-QL paper, as the original MGDT paper did not report MGDT-40M raw scores, while the Scaled-QL paper (by the same research team) did.
>
> The results for these 12 games show that:
>
> 1. The IQM HNS of MGDT-200M from our experiments and the original paper are 0.502 and 0.584 respectively.
> 2. The IQM HNS of MGDT-40M from our experiments and the original paper are 0.382 and 0.604 respectively.
> 3. Interestingly, the original MGDT-40M achieves higher IQM HNS than the original MGDT-200M on these 12 games (0.604 vs. 0.584).
>
> Given that the original MGDT was pretrained for 100M gradient steps while we pretrained all methods for only 1.75M steps, the IQM HNS difference for MGDT-200M is understandable. However, the surprisingly strong performance of the original MGDT-40M makes our reproduction results appear weaker.
>
> Table 2: Raw scores of MGDT from ours and MGDT's paper.
>
> | Game           | Random | Human   | Ours MGDT-40M | Original MGDT-40M | Ours MGDT-200M | Original MGDT-200M |
> | -------------- | ------ | ------- | ------------- | ----------------- | -------------- | ------------------ |
> | Assault        | 222.4  | 742.0   | 1227.2        | 1772.2            | 1741.5         | 2385.9             |
> | Atlantis       | 12850  | 29028.1 | 26657.1       | 304931.2          | 2565750.0      | 3105342.3          |
> | BeamRider      | 363.9  | 16926.5 | 972.0         | 3225.5            | 6011.3         | 8560.5             |
> | Carnival       | 0      | 3800    | 3460.0        | 3786.9            | 2610.0         | 2213.8             |
> | Centipede      | 2090.9 | 12017   | 3024.0        | 2867.5            | 4604.0         | 2463.0             |
> | ChopperCommand | 811.0  | 7387.8  | 2400.0        | 3337.5            | 3300.8         | 4268.8             |
> | DemonAttack    | 152.1  | 1971    | 1943.3        | 3629.4            | 6549.4         | 23768.4            |
> | NameThisGame   | 2292.3 | 8049    | 4691.4        | 7777.5            | 6610.5         | 9056.9             |
> | Phoenix        | 761.4  | 7242.6  | 3522.8        | 4744.4            | 5120.5         | 5295.6             |
> | Seaquest       | 68.4   | 42054.7 | 700.0         | 3112.5            | 2720.0         | 5173.8             |
> | TimePilot      | 3568   | 5229.2  | 4000.0        | 3487.5            | 3866.7         | 2743.8             |
> | Zaxxon         | 32.5   | 9173.3  | 125.0         | 4637.5            | 462.5          | 275.0              |
> | #IQM HNS       | 0.000  | 1.000   | 0.382         | 0.604             | 0.502          | 0.584              |
>
> ---
>
> > **[Q4]** Is the action-part module (CQL) hard to tune in this method?
>
> **[A7]** We appreciate your interest in the difficulty of tuing CQL. When combining the CQL term with distributional TD loss, we found the tuning of the CQL coefficient $\alpha$ to be relatively straightforward. We tested $\alpha$ values of 0.1 and 0.05, and both performed well. However, we encountered significant challenges when tuning $\alpha$ for the combination of CQL with MSE TD loss. As mentioned in our ablation study (line 489), we tested $\alpha$ values in {0.01, 0.05, 0.1}, but the agents consistently over-optimized the CQL term regardless of the $\alpha$ value. This led to extremely high Q-values for in-domain state-actions and low Q-values for OOD actions.

---

> ### Author Response · Authors · 2024-11-25
>
> Dear Reviewer pLGR,
>
> As the discussion deadline is approaching (<3 days), we are actively looking forward to your further feedback. Thanks for your effort and understanding!
>
> Kindest regards,
>
> Authors of ICLR Submission 1467

---

> ### Author Response · Authors · 2024-11-27
>
> Dear Reviewer pLGR,
>
> As the discussion deadline is approaching, we are actively looking forward to your valuable feedback and would be very grateful if you could take a moment to review our responses.
>
> We sincerely appreciate your precious time and consideration!
>
> Kindest regards,
>
> Authors of ICLR Submission 1467

---

> ### Author Response · Authors · 2024-11-29
>
> Dear Reviewer pLGR,
>
> First of all, we would like to wish you a Happy Thanksgiving! We hope you are enjoying this special holiday with your loved ones.
>
> As the discussion deadline is approaching, we are actively looking forward to your valuable feedback and would be very grateful if you could take a moment to review our responses when you have time after the holiday.
>
> We sincerely appreciate your precious time and consideration!
>
> Kindest regards,
>
> Authors of ICLR Submission 1467

---

> ### Author Response · Authors · 2024-12-02
>
> Dear Reviewer pLGR,
>
> As we draw closer to the discussion deadline (<1 day), we deeply value your expertise and perspective on our work, and we are eagerly anticipating your thoughtful feedback.
>
> Understanding that your schedule may be demanding, we would be immensely grateful if you could find a moment to share your insights before the upcoming deadline. Your contribution is crucial in helping us refine our research and enhance its impact within the academic community.
>
> Warmest regards,
>
> Authors of ICLR Submission 1467

---

> ### Author Response · Authors · 2024-12-03
>
> Dear Reviewer pLGR,
>
> As the rebuttal deadline is fast approaching (<2 hours), we kindly remind you that this is our final opportunity for discussion during rebuttal period. We would greatly appreciate it if you could spare a moment to review our responses. Your feedback is invaluable to us, and we sincerely hope for your consideration before the deadline.
>
> Thank you once again for your precious time and thoughtful evaluation.
>
> Kindest regards,
>
> Authors of ICLR Submission 1467

---

### Official Review · Reviewer_pKew · 2024-11-02

**Soundness:** 3
**Presentation:** 3
**Contribution:** 3
**Rating:** 6
**Confidence:** 4

**Summary:**

This paper introduces JOWA, a Jointly-Optimized World-Action model for offline RL. By optimizing a shared transformer backbone for world-action modeling and employing an efficient planning algorithm to improve policy search, JOWA achieves good performance on pretrained games with limited data and demonstrates superior generalization to new games with minimal fine-tuning trajectories.

**Strengths:**

1. This paper is clearly written and easy to follow.
2. This paper presents sufficient experimental results to demonstrate the validity of its proposed method.

**Weaknesses:**

1. A large generalist TD-MPC2 agent is capable of performing a variety of tasks across multiple domains. I wonder if the proposed method is better than TD-MPC2 in the offline setup.
2. Extending the experiments beyond Atari to more complex environments like Kitchen or Meta-World would offer stronger validation of the proposed method's effectiveness.

**Questions:**

Please see the weaknesses section.

---

> ### Author Response · Authors · 2024-11-21
> **Rebuttal by Authors [1/1]**
>
> Dear Reviewer pKew,
>
> We sincerely appreciate your valuable and insightful comments as they are extremely helpful for improving our manuscript. We have addressed each comment in detail in the paragraphs below.
>
> ---
>
> > **[w1]** A large generalist TD-MPC2 agent is capable of performing a variety of tasks across multiple domains. I wonder if the proposed method is better than TD-MPC2 in the offline setup.
>
> **[A1]** Thank you for pointing this out. Due to the inherent characteristics of the MPPI algorithm, TD-MPC series can't handle discrete control tasks, as explicitly stated in TD-MPC2's paper (see section I in its appendix). Similarly, the current version of JOWA can't handle continuous control tasks because it is as it is based on a C51-style architecture. Therefore, a direct comparison between these two methods is not currently feasible.
>
> ---
>
> > **[w2]** Extending the experiments beyond Atari to more complex environments like Kitchen or Meta-World would offer stronger validation of the proposed method's effectiveness.
>
> **[A2]** We appreciate your interset in extending JOWA to continous control tasks. We believe that extending JOWA to the continuous control tasks is an interesting and promising direction for future work. We have already started arranging computing resources for this project. Our ultimate goal is to use TD to learn a generalist/multi-task policy (on common benchmarks, such as Atari, ProcGen, DMLab, DMControl, and Meta-World) from the offline dataset while also focusing on efficient adaptation to OOD tasks. At that time, we will of course compare with TD-MPC2 on continuous control tasks.
>
> We sincerely apologize that due to the time limit of the rebuttal period, we have allocated our resources to other experiments, making it impossible to demonstrate JOWA's performance on continuous tasks at present. However, we can outline 3 potential approaches for extending JOWA to continuous environments:
>
> 1. C51-style approach: We divide the value range of each action dimension evenly into $K$ intervals and use the mean of the interval to represent the value of this interval. We then form the discrete action space by taking the union of all possible discrete values and treat it as a discrete control task using C51-style method to train an **action-level optimal Q-function**. While this approach requires minimal code changes, we have concerns about the performance impact of discretization and the exponential growth of the action space in high-dimensional environments.
> 2. Q-Transformer-style approach: After discretizing each action dimension, we learn a **dimension-level (rather than action-level) optimal Q-function** like Q-Transformer[1]. This approach requires moderate code modifications. However, we are concerned that the additional Bellman updates across action dimensions might exacerbate the credit assignment problem in high-dimensional action tasks.
> 3. REINFORCE-style approach: Following action dimension discretization, we learn an action-level Q-function (not optimal Q-function) where the Q-head takes the last action dimension token's embedding as input. We calculate the target Q-value using the Monte Carlo method with the corresponding offline trajectory and treat Q-head training as a multi-class classification problem using CELoss (with CQL penalty) for stability. Then A dimension-level policy head uses each one of embeddings from last observation token to the penultimate action dimension token to predict next action dimension, trained via REINFORCE rather than auto-regression. This framework (**action-level Q-function with dimension-level policy**) seems somehow like ArCHer[2], but we use the shared backbone for Q-network and policy network and also incorporates world model for planning. This method requires substantial code changes and potential re-experimentation on Atari, but in my experience it may be the most stable way for training.
>
> Currently, we have begun preparing computing resources for experiments on small-scale multi-continuous tasks. We plan to start with the second method as we progress toward our ultimate goal.
>
> [1] Chebotar, Yevgen, et al. "Q-transformer: Scalable offline reinforcement learning via autoregressive q-functions." *Conference on Robot Learning*. PMLR, 2023.
>
> [2] Zhou, Yifei, et al. "Archer: Training language model agents via hierarchical multi-turn rl." *arXiv preprint arXiv:2402.19446* (2024).

---

> ### Author Response · Authors · 2024-11-25
>
> Dear Reviewer pKew,
>
> As the discussion deadline is approaching (<3 days), we are actively looking forward to your further feedback. Thanks for your effort and understanding!
>
> Kindest regards,
>
> Authors of ICLR Submission 1467

---

> > ### Comment · Reviewer_pKew · 2024-11-26
> >
> > I appreciate the response provided by the author. I increase my confidence by 1.

---

> > > ### Author Response · Authors · 2024-11-26
> > >
> > > Thank you for recognizing our work and for your time and effort as a reviewer!

---

### Official Review · Reviewer_7Vxq · 2024-11-04

**Soundness:** 4
**Presentation:** 3
**Contribution:** 3
**Rating:** 8
**Confidence:** 4

**Summary:**

This paper introduces a model-based RL agent, JOWA, which employs a transformer architecture and jointly optimizes world dynamics and Q-values across different environments. To compensate for inaccurate Q-values, the learned world model enables JOWA to search out the optimal policy via planning during inference time. Empirically, JOWA demonstrates impressive performance in a low-data regime by training on 15 Atari games. Moreover, JOWA's performance scales up with model sizes.

**Strengths:**

- The proposed JOWA outperforms existing SOTA methods by a large margin. The perform scales up with model sizes

- The joint optimization of world and action models stabilizes large-scale multi-task offline RL training.

- The ablation studies comprehensively study the key design choices of the proposed methods.

**Weaknesses:**

- The proposed method combines the best offline RL training techniques, leveraging the world modeling loss to stabilize Q-value learning. The empirical performance is impressive. However, the technical novelty is thus limited.

- By taking a closer look at Table 2, we can see that the 150M variant does not consistently outperform the 40M and 70M variants on all tasks. For example, the 40M variant achieves the highest score on Centipede, while the 70M variant excels on NameThisGame, SpaceInvaders, and StarGunner. Can you explain the phenomenon a bit more? Is it because of the training instability? Can further increase the model size and training iterations solve the issue?

- It would be insightful to examine the emergent behaviors that arise when scaling up model size. For example, the 150M JOWA achieves a significantly higher score than the 70M JOWA and the other baseline methods on *Zaxxon*. Does the 150M JOWA agent exhibit any distinct emergent behaviors that could explain this performance improvement?

- This work misses the citation for [1,2]

[1] Li et al., Multi-task batch reinforcement learning with metric learning. NeurIPS 2020.

[2] Li et al., Offline reinforcement learning with closed-form policy improvement operators. ICML 2023

**Questions:**

How's the model performance when you further scale up the model size, e.g., to 1B?

---

> ### Author Response · Authors · 2024-11-21
> **Rebuttal by Authors [1/1]**
>
> Dear Reviewer 7Vxq,
>
> We sincerely appreciate your valuable and insightful comments as they are extremely helpful for improving our manuscript. We have addressed each comment in detail in the paragraphs below.
>
> ---
>
> > **[w1]** The proposed method combines the best offline RL training techniques, leveraging the world modeling loss to stabilize Q-value learning. The empirical performance is impressive. However, the technical novelty is thus limited.
>
> **[A1]** Thank you for pointing this out. We think that the novelty of JOWA can be examined in the following ways:
>
> 1. JOWA examines which existing technologies are crucial for scaling RL, providing a novel perspective on evaluating established techniques.
> 2. JOWA introduces a novel planning algorithm. Our ablation experiments (Table 4) demonstrate that this planning algorithm significantly improves performance. As detailed in our response **[A5]** to reviewer zGRd, when compared to MCTS, our algorithm is not only 10$\times$ faster but also achieves better performance.
>
> ---
>
> > **[w2]** By taking a closer look at Table 2, we can see that the 150M variant does not consistently outperform the 40M and 70M variants on all tasks. For example, the 40M variant achieves the highest score on Centipede, while the 70M variant excels on NameThisGame, SpaceInvaders, and StarGunner. Can you explain the phenomenon a bit more? Is it because of the training instability? Can further increase the model size and training iterations solve the issue?
>
> **[A2]** Thank you for highlighting this. We checked the action sequence output by JOWA-40M on Centipede and found that the agent consistently executes the `rightfire` action after the first few steps, which is not a fully meaningful behavior but surprisingly dodge enemy attacks and get the highest score. We further evaluated the last 5 checkpoints of JOWA-70M and JOWA-150M and found the winner on NameThisGame, SpaceInvaders, and StarGunner to be irregular, i.e., no one wins each of the game consistently.
>
> Moreover, upon careful examination of table C.2 in the appendix of Scaled-QL's paper[1], we found that MGDT-200M also performs worse than MGDT-40M on 13 games (BankHeist, Carnival, Centipede, FishingDerby, Freeway, Gravitar, IceHockey, Jamesbond, KungFuMaster, TimePilot, VideoPinball, WizardOfWor, and Zaxxon). Therefore, we believe this inconsistence is an inherent problem of offline RL and unfortunately, neither increasing model size (such as scaling from 40M to 200M) nor extending training iterations (even though the original MGDT[2] was trained for 100M gradient steps) can solve this issue.
>
> [1] Kumar, Aviral, et al. "Offline q-learning on diverse multi-task data both scales and generalizes." *arXiv preprint arXiv:2211.15144* (2022).
>
> [2] Lee, Kuang-Huei, et al. "Multi-game decision transformers." *Advances in Neural Information Processing Systems* 35 (2022): 27921-27936.
>
> ---
>
> > **[w3]** It would be insightful to examine the emergent behaviors that arise when scaling up model size. For example, the 150M JOWA achieves a significantly higher score than the 70M JOWA and the other baseline methods on *Zaxxon*. Does the 150M JOWA agent exhibit any distinct emergent behaviors that could explain this performance improvement?
>
> **[A3]** Thank you for pointing this out. We examined the reward of Zaxxon game and found that this "emergent behaviors" is actually attributable to the nonlinear reward function. The first reward in Zaxxon is often nearly 5000. JOWA-40M gets the large reward in one of the 16 rollouts while JOWA-150M gets it in 7 rollouts. In other words, when we clip the rewards to the range [-1,1], JOWA-70M and JOWA-150M achieve scores of 0.11 and 0.44 respectively, indicating no true 'emergence' phenomenon. We find the literature[1] draws a similar conclusion that nonlinear or discontinuous metrics cause the emergent abilities.
>
> [1] Schaeffer, Rylan, Brando Miranda, and Sanmi Koyejo. "Are emergent abilities of large language models a mirage?." *Advances in Neural Information Processing Systems* 36 (2024).
>
> ---
>
> > **[w4]** This work misses the citation for [1,2]
>
> **[A4]** Thank you for your suggestion. We will add these two works to our related work section to improve our paper.
>
> ---
>
> > **[Q1]** How's the model performance when you further scale up the model size, e.g., to 1B?
>
> **[A5]** We appreciate your interest in further scaling the model size. While this is an interesting and important question, due to our current computational constraints, we cannot complete such experiments during the rebuttal period. We estimate that training JOWA-1B would require 16 A100 GPUs running for 1-2 months. Therefore, we will leave this as future work.

---

> > ### Comment · Reviewer_7Vxq · 2024-11-24
> > **Thank you for your response!**
> >
> > Thank you for your detailed response. I increase my confidence by 1.
> >
> > I suggest you highlight the efficiency of your planning algorithm in your revised manuscript.
> >
> > I like your responses in A2 and A3, and I suggest you discuss the actual behavior of the agent in the Appendix. Moreover, please include the phenomenon of "inversion scaling law", i.e., increasing model parameters leads to decreasing performance in specific games, in the limitation section, in the Limitation section.
> >
> > Lastly, I would like to remind the authors that they can update the manuscript during the rebuttal period to incorporate their modifications. Doing so could help address, for example, Reviewer zGRd's concerns more effectively.

---

> > > ### Author Response · Authors · 2024-11-25
> > >
> > > Thank you for recognizing our work and for your time and effort as a reviewer! We will upload an updated manuscript within the next 24 hours to include new experiments, conclusions, related work, limitations, etc. added during the rebuttal period.

---

### Official Review · Reviewer_zGRd · 2024-11-05

**Soundness:** 3
**Presentation:** 3
**Contribution:** 2
**Rating:** 5
**Confidence:** 4

**Summary:**

This paper presents JOWA, an offline model-based reinforcement learning method aimed at scaling and generalizing multi-task RL through a shared transformer-based architecture. JOWA is designed to stabilize temporal difference learning for large models by integrating world modeling with Q-value criticism, thus leveraging a shared transformer backbone for both tasks. The paper introduces a novel parallelizable planning algorithm to counter Q-value estimation errors, achieving more consistent policy identification during inference. The model is pre-trained on Atari games with minimal data and reportedly outperforms existing baselines, demonstrating some generalization capacity to new tasks.

**Strengths:**

1. Scalability: JOWA’s design showcases robust scaling potential, as performance improves with model size without the usual TD-learning instability issues.

2. Detailed Ablation Studies: The authors conducted extensive ablations, examining the impact of core design elements such as task embeddings, training losses, and synthetic data usage.

**Weaknesses:**

1. Missing related work and explanations for the architecture of the world-action modeling. The work proposes to use VQ-VAE for the representation learning in Atari games, while there is a work named Forward-Inverse Cycle Consistency (FICC), which uses VQ-VAE for the offline Atari dataset to learn representations and action embeddings. It seems this pipeline is so similar to the FICC, however, the author didn't mention the difference between JOWA and FICC. (Both use offline Atari datasets for model-based RL and learning universal policy under VQ-VAE loss terms)

Ye, W., Zhang, Y., Abbeel, P., & Gao, Y. (2022). Become a proficient player with limited data through watching pure videos. In The Eleventh International Conference on Learning Representations.

2. Overclaim of the generalization ability. The experiments are based on Atari games, while using offline atari dataset for pre-training. Extending this approach to more complex control task scenarios (e.g., robotics or high-dimensional continuous control tasks) remains unproven. I'm not sure about the effectiveness of the scaling results.

**Questions:**

1. How might JOWA perform in a single-task setting compared to standard model-based methods? Would the shared architecture yield efficiency or performance gains? If it is worse than training from scratch with model-based RL, why the Atari games are good benchmarks for evaluating the scaling laws?

2. The author mentioned the beam search, so why not use MCTS, which is proven effective in MuZero and EfficientZero. And the author didn't compare with other model-based RL algorithms, such as Dreamer.

---

> ### Author Response · Authors · 2024-11-21
> **Rebuttal by Authors [1/5]**
>
> Dear Reviewer zGRd,
>
> We sincerely appreciate your valuable and insightful comments as they are extremely helpful for improving our manuscript. We have addressed each comment in detail in the paragraphs below.
>
> ---
>
> > **[w1]** Missing related work and explanations of FICC.
>
> **[A1].** Thank you for bringing this to our attention. We apologize for missing this excellent paper due to our limited knowledge. We have modified our paper to include FICC as related work and have employed it as a model-based baseline for comparison (see **[A4]**). We explain the differences between JOWA and FICC as follows:
>
> 1. FICC pretrains representation and dynamic networks using action-free videos and then primarily fine-tunes reward, value and policy networks on downstream tasks with an action adapter, while JOWA pretrains all networks (representation, dynamic, reward, value networks) with action-aware trajectories. Actually, this is the main difference between FICC and JOWA. FICC and JOWA represent two different pretraining objectives, i.e., pretraining with videos or trajectory data respectively. This main difference affects both in-domain performance and OOD fine-tuning performance (see **[A4]**).
> 2. FICC pretrains the inverse dynamic model, forward dynamic model, and latent action codebooks in a VQ-VAE style pipeline, while JOWA only employs VQ-VAE as the image tokenizer and pretrains the dynamic model in the auto-regression way.
> 3. The Original FICC obtains a multi-task dynamic model while gets a single-task policy, as shown in section 5.3 of their paper, the authors pretrained the representation network and dynamic model on 6 environments while fine-tuned on each environment respectively. In contrast, JOWA obtains a multi-task policy after pretraining. But we can build action adapters for each environment independently and then aggregate them into a two-level dictionary for multi-task fine-tuning of FICC, so this is not a big difference.
> 4. FICC only open sources the pretraining codes, no fine-tuning code and model weights, while JOWA open source pretraining, fine-tuning, evaluation codes, and model weights. Due to the extensive training time, we believe that our fully open-source approach was necessary to provide the community with valuable sources.
>
> In conclusion, we thanks you again for bringing this excellent work to us. The differences and performance comparison between FICC and JOWA helps us to improve our paper and gain deeper insights into the impact of different objectives on RL pre-training.
>
> ---
>
> > **[w2]** Overclaim of the generalization ability.
>
> **[A2]** Thank you for highlighting this. However, we have carefully reviewed our contribution on generalization in abstract and introduction, and believe that we have clearly restricted our conclusions about generalization to the Atari game domain. For example, we state that "JOWA ... and can sample-efficiently transfer to novel games using only 5k offline fine-tuning data (approximately 4 trajectories) per game, demonstrating superior generalization." in abstract and "Third, JOWA enables sample-efficient transfer to diverse unseen games with 64.7% DQN-normalized score using only 5k transitions per game, surpassing baselines by 34.7% on average." in introduction. Following MGDT and Scaled-QL, our work focuses on Atari games and considers generalization to OOD Atari games. Therefore, we cautiously state our conclusions about generalization, limiting the property to the Atari domain.
>
> We believe that extending JOWA to the continuous control tasks is an interesting and promising direction for future work. We have already started arranging computing resources for this project. Our ultimate goal is to use TD to learn a generalist/multi-task policy (on common benchmarks, such as Atari, ProcGen, DMLab, DMControl, and Meta-World) from the offline dataset while also focusing on efficient adaptation to OOD tasks.

---

> ### Author Response · Authors · 2024-11-21
> **Rebuttal by Authors [2/5]**
>
> > **[Q1]** JOWA vs. standard model-based methods in a single-task setting.
>
> **[A3]** Thank you for pointing this out. We would highlight that our motivation is building a multi-task policy on Atari from the offline dataset. Therefore, we focus on validating the efficiency and performance gains of the shared architecture under multi-task pretraining setting, and don't compare with common online model-based RL algorithms, such as IRIS, STORM, Muzero, Efficient zero et al., on Atari 100k benchmark, which is beyond the scope of this work. However, our fine-tuning experiments are in the single-task setting, i.e., we fine-tune pretrained models on each OOD game respectively. Here, we implement 2 more offline model-based RL algorithms: MOReL[1] and COMBO[2], which are trained from scratch with 5k transitions along with JOWA-150M trained from scratch and pretrained models. MOReL constructs a conservative MDP with ensembled dynamic models and penalizes on reward according to dynamic uncertainty. COMBO implemented here detaches the Q-head from transformer backbone, resulting in individual optimization of Q function and dynamic model. We show the results in the following table. The results show that:
>
> 1. The performance gains from JOWA's designs (such as shared backbone, planning et al.) have limited benefits when training from scratch in a offline single-task setting, compared with other offline model-based RL algorithms.
>
> 2. Pretrained models still demonstrate superior fine-tuning benefits over methods trained from scratch.
>
> 3. JOWA's designs mainly improve pretraining performance straightforwardly, which in turn facilitates fine-tuning performance and helps pretrained JOWA-150M obtains the best fine-tuning performance in our experiments.
>
> We would highlight that a choice is ordinary in the single-task setting while important in the multi-task setting does not indicate the choice is useless. For example, in the single-task setting, both mean-squared TD error and distributional TD error perform comparably online [3] and offline [4,5]. However, some works [6,7] and we observe that MSE TD error does not scale well under multi-task setting, and performs much worse than distributional TD error (shown in our ablation study).
>
> Table 1: Results of the single-task fine-tuning experiment.
>
> |        | MTBC-120M | MGDT-200M | EDT-200M | SQL-80M | JOWA-150M | JOWA-150M (scratch) | MoReL-130M (scratch) | COMBO-150M (scratch) |
> | ------ | --------- | --------- | -------- | ------- | --------- | ------------------- | -------------------- | -------------------- |
> | Mean   | 0.164     | 0.422     | 0.430    | 0.360   | 0.647     | 0.196               | 0.207                | 0.168                |
> | Median | 0.215     | 0.354     | 0.325    | 0.284   | 0.615     | 0.173               | 0.168                | 0.173                |
> | IQM    | 0.205     | 0.377     | 0.380    | 0.355   | 0.647     | 0.181               | 0.214                | 0.153                |
>
> [1] Kidambi, Rahul, et al. "Morel: Model-based offline reinforcement learning." *Advances in neural information processing systems* 33 (2020): 21810-21823.
>
> [2] Yu, Tianhe, et al. "Combo: Conservative offline model-based policy optimization." *Advances in neural information processing systems* 34 (2021): 28954-28967.
>
> [3] Agarwal, Rishabh, et al. "Deep reinforcement learning at the edge of the statistical precipice." *Advances in neural information processing systems* 34 (2021): 29304-29320.
>
> [4] Kumar, Aviral, et al. "Conservative q-learning for offline reinforcement learning." *Advances in Neural Information Processing Systems* 33 (2020): 1179-1191.
>
> [5] Kumar, Aviral, et al. "Dr3: Value-based deep reinforcement learning requires explicit regularization." *arXiv preprint arXiv:2112.04716* (2021).
>
> [6] Kumar, Aviral, et al. "Offline q-learning on diverse multi-task data both scales and generalizes." *arXiv preprint arXiv:2211.15144* (2022).
>
> [7] Farebrother, Jesse, et al. "Stop regressing: Training value functions via classification for scalable deep rl." *arXiv preprint arXiv:2403.03950* (2024).

---

> ### Author Response · Authors · 2024-11-21
> **Rebuttal by Authors [3/5]**
>
> > **[Q2]** Why the Atari games are good benchmarks for evaluating the scaling laws?
>
> **[A4]** Thank you for bringing this to our attention. We chose Atari as the benchmark for evaluating scaling laws for the following reasons:
>
> 1. Inspired by generative world models like SORA, our motivation is to build a large image observation-based world model from the offline dataset. Therefore, we focus on offline vision tasks.
> 2. We investigate the offline RL datasets shown in the following table. After comprehensively consider the vision-task constraint and whether the data volume is sufficient, we finally choose Atari as the benchmark.
> 3. We investigate the works about pretraining using Atari dataset and find MGDT, EDT and Scaled-QL, which provide sound and reproducible baselines.
> 4. Atari is one of the most common and reliable benchmarks in the development of RL in the past decade. Atari has various types of games, each with different embodiments, dynamics, and reward functions, which makes Atari more challenging than other multi-task benchmarks such as Meta-World.
> 5. The performance of multi-task policy on Atari is still behind that of single-task policy, which means there exists room for improvement for multi-task policy on Atari. For example, MGDT shows that the MGDT-200M achieves 126% human-normalized scores (HNS) on 40 Atari games while single-task offline BCQ and single-task online DQN achieves 135% and 144% HNS.
>
> Table 2: Offline RL datasets.
>
> | Datasets       | Vision/State | Continous/Discrete | Data volume                                                  | Links                                                        |
> | -------------- | ------------ | ------------------ | ------------------------------------------------------------ | ------------------------------------------------------------ |
> | D4RL           | State        | Continous          | about 1M transitions per task                                | [repo](https://github.com/Farama-Foundation/D4RL/tree/master) |
> | Atari          | Vision       | Discrete           | 50M*5 per game, 60 games in total                            | [repo](https://github.com/google-research/batch_rl) / [wrapper](https://github.com/takuseno/d4rl-atari) |
> | v-d4rl         | Vision       | Continous          | about 100k*5 per task, 3 tasks in total                      | [repo](https://github.com/conglu1997/v-d4rl)                 |
> | Robomimic      | Hybrid       | Continous          | Lift Real (PH) 1.9G/Can Real (PH) 5.3G/Tool Hang Real (PH) 58G | [website](https://robomimic.github.io/docs/datasets/robomimic_v0.1.html) |
> | Mimicgen       | Hybrid       | Continous          | 12 tasks, 48k demos                                          | [website](https://robomimic.github.io/docs/datasets/mimicgen.html) |
> | RoboTurk Pilot | Hybrid       | Continous          | 1k demos                                                     | [website](https://robomimic.github.io/docs/datasets/roboturk_pilot.html) |

---

> ### Author Response · Authors · 2024-11-21
> **Rebuttal by Authors [4/5]**
>
> > **[Q3]** Why not use MCTS?
>
> **[A5]** We appreciate your interest in using MCTS as the planning algorithm. Actually, we conducted a preliminary experiment to evaluate the effectiveness of MCTS and found that MCTS is about 10x slower than our proposed planning algorithm due to non-parallelizability of MCTS. Here, to answer this question more formally, we conducted a more comprehensive experiment on planning JOWA with MCTS. We implement muzero-style MCTS in python rather than C++ for fair speed comparison. With the same expansion state budget, our beam search style planning and MCTS achieves 1.26FPS and 0.12FPS respectively.
>
> For MCTS, we conduct a grid search on the following choices:
>
> 1. Compute V-value using `Q.mean()` or `Q.max()` (note that JOWA only pretrains a optimal Q-function, thus no V-function and policy network available).
>
> 2. We employed an energy-based policy to compute action probability, i.e., $\pi=softmax(\frac{Q}{t})$, where $t$ is the temperature. We search $t$ in {0.01, 0.1, 0.5, 0.7, 0.8, 0.9, 1, 2, 3, 5, 10}.
>
> 3. Use most visited or most valuable action as the optimal action, i.e., `argmax(root.children.visit_count)` or `argmax(root.children.value)`.
>
> 4. Search the max depth of the tree $H$ in [1,7].
>
> Finally, we selected the configuration using `Q.max()`, $t=0.9$, and `argmax(root.children.value)` for all games while searching $H$ in {1, 2, 4, 6} for each game. Even though we believe we did a sufficiently adequate hyperparameter search, MCTS still performs worse than beam search, as shown in the following table.
>
> The results show that beam search exceeds MCTS by 71.0% Mean HNS. Moreover, we find that MCTS is highly sensitive to the temperature $t$ and max depth $H$, and inappropriate values of hyperparameters can even degenerate the policy into a randomized policy. However, its long execution time makes it inconvenient to tune hyperparameters.
>
> Table 3: MCTS vs. beam search on JOWA-150M
>
> | Game           | Random | Human   | JOWA w/o planning | JOWA with MCTS | JOWA with beam search |
> | -------------- | ------ | ------- | ----------------- | -------------- | --------------------- |
> | BeamRider      | 363.9  | 16926.5 | 864.5             | 1137.0         | 3498                  |
> | Berzerk        | 123.7  | 2630.4  | 396.9             | 440.0          | 739                   |
> | Carnival       | 0      | 3800    | 5560.0            | 3340.0         | 5316                  |
> | ChopperCommand | 811.0  | 7387.8  | 806.2             | 1850.0         | 3812.5                |
> | Seaquest       | 68.4   | 42054.7 | 267.5             | 760.0          | 2725                  |
> | TimePilot      | 3568   | 5229.2  | 662.5             | 4100.0         | 3669                  |
> | Zaxxon         | 32.5   | 9173.3  | 12.5              | 50.0           | 2163                  |
> | #Mean HNS      | 0.000  | 1.000   | -0.02             | 0.221          | 0.378                 |
> | IQM HNS        | 0.000  | 1.000   | 0.03              | 0.134          | 0.237                 |
>
> ---
>
> > **[Q4]** JOWA didn't compare with other model-based RL algorithms, such as Dreamer.
>
> **[A6]** Thank you for raising this point. We didn't compare JOWA with other online model-based RL baselines such as Dreamer for the following reasons:
>
> 1. Dreamer is proposed under online setting, while our work focuses on offline RL. Thus for fair comparison, we need at least adding CQL regularization for Dreamer.
> 2. Dreamer needs training with imagination trajectories and uses a high update-to-data (UTD) ratio of 512, both of which are time-consuming and would take more than 6 months to pretrain on 15 games with 16*A100 GPUS. Thus, we need disable model-based data synthesis to make the training time acceptable, just like JOWA's pretraining.
> 3. Since we disable the ability of world model during pretraining, we need to use the world model after pre-training, e.g. planning with the world model.
> 4. After the above 3 changes to Dreamer, the differences between the modified Dreamer and JOWA are not obvious. The main difference left is the shared backbone of JOWA, which has been proven effective in ablation experiments.
>
> However, we believe that the FICC you mentioned is a worthwhile baseline for comparison. As we stated in **[A1]**, FICC and JOWA represent two different pre-training objectives, so it is well worth discussing the impact of these two objectives on the in-domain and OOD Atari games.
>
> [Part 1 of 2 for **[A6]**, to be continued ...]

---

> ### Author Response · Authors · 2024-11-21
> **Rebuttal by Authors [5/5]**
>
> [Part 2 of 2 for **[A6]**]
>
> We implement FICC-L through replacing the residual blocks in the representation, dynamic, and LAG networks with ResNet-50 style residual blocks, resulting in FICC-85M. We use the same 10% down-sampled dataset for pretraining representation and dynamic models for 0.5M steps, computing action adapter, and primarily fine-tuning Q-function on 15 games all at once for 1.25M steps. To save time and ensure fair comparison on pretraining objectives, we employ the same training method of Q-function as JOWA when fine-tuning FICC and evaluate FICC-85M with beam search planning, rather than fine-tuning with EfficientZero. We use 32*A100 GPUs to train FICC-85M for 5 days. Here, we show the results of FICC-85M on both in-domain 15 games and OOD 5 games in the following tables. These results show that:
>
> 1. JOWA-70M and JOWA-150M exceeds FICC-85M 9.9% and 82.2% IQM HNS on in-domain 15 Atari games, respectively.
> 2. JOWA-70M and JOWA-150M exceeds FICC-85M 4.9% and 12.5% IQM DNS on OOD 5 Atari games, respectively.
>
> Therefore, we conclude that within the Atari domain, JOWA's objective is better than FICC. We find that the literature[1] draws a similar conclusion that pretraining with trajectory data is better than video data on Atari domain (see Figure 2 in [1], where ID and Near-OOD are both Atari games).
>
> We will include all the experiments and discussions about FICC in our main paper. Thank you again for bringing this excellent work to us, which helps us to improve our paper and gain deeper insights into the impact of different objectives on RL pre-training.
>
> [1] Kim, Donghu, et al. "Investigating Pre-Training Objectives for Generalization in Vision-Based Reinforcement Learning." *arXiv preprint arXiv:2406.06037* (2024).
>
> Table 4: FICC vs. JOWA on in-domain Atari games
>
> | Game           | Random | Human   | FICC-85M | JOWA-70M | JOWA-150M |
> | -------------- | ------ | ------- | -------- | -------- | --------- |
> | Assault        | 222.4  | 742.0   | 925.9    | 1733.9   | 2302      |
> | Atlantis       | 12850  | 29028.1 | 86250    | 570862.5 | 2690387.5 |
> | BeamRider      | 363.9  | 16926.5 | 6822     | 2547.4   | 3498      |
> | Berzerk        | 123.7  | 2630.4  | 400      | 441.9    | 739       |
> | Carnival       | 0      | 3800    | 2820     | 4070     | 5316      |
> | Centipede      | 2090.9 | 12017   | 3742.2   | 4475.6   | 4677      |
> | ChopperCommand | 811.0  | 7387.8  | 2835.6   | 2568.8   | 3812.5    |
> | DemonAttack    | 152.1  | 1971    | 5806.4   | 4584.4   | 3547.8    |
> | NameThisGame   | 2292.3 | 8049    | 6236     | 12706.9  | 11421     |
> | Phoenix        | 761.4  | 7242.6  | 3814.5   | 5065     | 5348      |
> | Seaquest       | 68.4   | 42054.7 | 1760     | 1490     | 2725      |
> | SpaceInvaders  | 148    | 1668.7  | 641.2    | 969.1    | 744.7     |
> | StarGunner     | 664.0  | 10250   | 4936.4   | 21231.3  | 18150     |
> | TimePilot      | 3568   | 5229.2  | 4166.7   | 3831.3   | 3669      |
> | Zaxxon         | 32.5   | 9173.3  | 312.5    | 225      | 2163      |
> | #IQM HNS       | 0.000  | 1.000   | 0.433    | 0.476    | 0.789     |
>
> Table 5: FICC vs. JOWA on OOD Atari games
>
> | Game        | Random | Human   | FICC-85M | JOWA-70M | JOWA-150M |
> | ----------- | ------ | ------- | -------- | -------- | --------- |
> | Gravitar    | 173.0  | 473.0   | 342.5    | 387.4    | 273.3     |
> | MsPacman    | 307.3  | 3085.6  | 1252.2   | 908.0    | 2016.7    |
> | Pong        | -20.7  | 19.5    | 14.0     | 14.6     | 17.7      |
> | Robotank    | 2.2    | 63.9    | 11.5     | 13.8     | 25.0      |
> | YarsRevenge | 3092.9 | 18089.9 | 15036.2  | 16339.2  | 17506.2   |
> | #Mean DNS   | 0.000  | 1.000   | 0.543    | 0.576    | 0.647     |
> | Median DNS  | 0.000  | 1.000   | 0.565    | 0.715    | 0.615     |
> | IQM DNS     | 0.000  | 1.000   | 0.575    | 0.603    | 0.647     |

---

> ### Author Response · Authors · 2024-11-25
>
> Dear Reviewer zGRd,
>
> As the discussion deadline is approaching (<3 days), we are actively looking forward to your further feedback. Thanks for your effort and understanding!
>
> Kindest regards,
>
> Authors of ICLR Submission 1467

---

> ### Author Response · Authors · 2024-11-27
>
> Dear Reviewer zGRd,
>
> As the discussion deadline is approaching, we are actively looking forward to your valuable feedback and would be very grateful if you could take a moment to review our responses.
>
> We sincerely appreciate your precious time and consideration!
>
> Kindest regards,
>
> Authors of ICLR Submission 1467

---

> > ### Comment · Reviewer_zGRd · 2024-11-27
> > **Official Comment by Reviewer zGRd**
> >
> > Thank you for your clarification. I like your detailed results and I understand the difference between JOWA and FICC.
> > However, I am not convinced by the experiments on Atari to evaluate the scaling laws. Like the Sora you've mentioned, it is trained toward real-world prediction. I guess you want to mention Genie or sth like that. But they have no such claim.
> >
> > Nevertheless, I have raised my score from 3 to 5.

---

> > > ### Author Response · Authors · 2024-11-27
> > >
> > > Thank you for your valuable feedback and for raising score. We truly appreciate your time and effort as a reviewer!
> > >
> > > As researchers in RL, we would be incredibly grateful if you could share your insights on why Atari might not be suitable for verifying RL scaling laws, and what benchmarks you would consider more appropriate. While we fully understand this goes beyond typical review responsibilities, your expert perspective would be invaluable for guiding our subsequent series of work in this area.
> > >
> > > We genuinely eager to learn from your expertise and would deeply appreciate any thoughts you're willing to share, as they would help us better understand this field.

---

### Author Response · Authors · 2024-11-25
**Update manuscript**

We thank reviewers for all the valuable feedback. We address all the reviewers' comments below and have incorporated all feedback in the revised manuscript  using blue font. Specifically, we have updated the following:
1. Add several relevant works in section 2 (Reviewer zGRd, Reviewer 7Vxq).
2. Employ FICC as a model-based baseline for main experiments in section 5 (Reviewer zGRd).
3. Add JOWA's scaling results and conclusions in the fine-tuning experiments, i.e., section 5.5 (Reviewer XU9g).
4. Add a comparison with MCTS in section 5.6 to emphasize the efficiency of our planning algorithm (Reviewer zGRd).
5. Add a description of the scaling inconsistency in the limitations (Reviewer 7Vxq).
6. Add additional experiments in Appendix F, including details of MCTS (Reviewer zGRd), fine-tuning results using non-expert data (Reviewer pLGR), and explanations of the emergent behaviors (Reviewer 7Vxq).

We sincerely aspire that our detailed rebuttal will dispel any uncertainties or misunderstandings which reviewers may have raised regarding our manuscript, thus contributing positively to the final ratings of this work. If any additional experiments are needed to further demonstrate the potential of JOWA, we will do our utmost to supplement the relevant experiments during the valuable discussion period.

---

### Author Response · Authors · 2024-12-04
**Summary of Rebuttal**

We are deeply grateful to all reviewers for their thorough evaluation of our work and their constructive feedback. We particularly appreciate their recognition of our paper's key strengths:

1. Strong Baseline Performance: Reviewers 7Vxq, pLGR, and XU9g
2. Scalability: Reviewers zGRd, 7Vxq, pLGR, and XU9g
3. Adaptability: Reviewer pLGR
4. Comprehensive Ablation Studies: Reviewers zGRd, 7Vxq, and pKew
5. Open-Source codes: Reviewer XU9g

Through productive discussions during the rebuttal period, we have made substantial improvements to our paper to address the reviewers' concerns:

1. Added additional relevant works (Reviewers zGRd, 7Vxq)
2. Added a model-based baseline for main experiments (Reviewer zGRd)
3. Added JOWA's scaling results and conclusions in fine-tuning experiments (Reviewer XU9g)
4. Added a comparison between our planning algorithm and MCTS (Reviewer zGRd)
5. Added a discussion of scaling inconsistency in the limitations section (Reviewer 7Vxq)
6. Added fine-tuning results using non-expert data (Reviewer pLGR)
7. Added the explanation of emergent behaviors (Reviewer 7Vxq)

After rebuttal, the main remaining concern from reviewers is that Atari might not be an ideal benchmark for evaluating RL's scaling law (Reviewer zGRd). We will carefully consider the reviewers' suggestions and design experiments based on their comments in our subsequent series of work, such as expanding to continuous action spaces.

We extend our sincere gratitude to the reviewers for their valuable insights, and to the Area Chair, Senior Area Chair, and Program Committee for their dedication in organizing this review process!

---

### Meta-Review · Area_Chair_NdZV · 2024-12-23

**Metareview:**

This paper introduces a model-based RL agent, JOWA, which trains a single transformer architecture by jointly optimizing the world dynamics and Q-values across different environments. To compensate for inaccurate Q-values, the learned world model enables JOWA to search out the optimal policy via planning as well, and JOWA can be fine-tuned online and offline as well.

The reviewers generally liked the paper and the only concern was about training on Atari to plot scaling laws. There's prior work which uses multi-game Atari to evaluate scaling laws, and while I think this is not the end goal for the RL community (and indeed, this paper does only consider 15 Atari games), this work is a good starting point for scaling of model-based RL.

I am also wondering if comparisons to TD-MPC2 could be added to the mix as well (I understand it cannot be applied here directly, but with approximations with action and/or state discretization), since that's a model-based method as well which shows scaling curves, although on different domains. Nonetheless, the paper is of interest and value, and we are accepting this paper.

**Additional Comments On Reviewer Discussion:**

See above.

---

### Decision · Program_Chairs · 2025-01-22

Accept (Poster)